# Diffusion Normalizing Flow

**Qinsheng Zhang**
Georgia Institute of Technology
qzhang419@gatech.edu

**Yongxin Chen**
Georgia Institute of Technology
yongchen@gatech.edu

## Abstract

We present a novel generative modeling method called diffusion normalizing flow based on stochastic differential equations (SDEs). The algorithm consists of two neural SDEs: a forward SDE that gradually adds noise to the data to transform the data into Gaussian random noise, and a backward SDE that gradually removes the noise to sample from the data distribution. By jointly training the two neural SDEs to minimize a common cost function that quantifies the difference between the two, the backward SDE converges to a diffusion process the starts with a Gaussian distribution and ends with the desired data distribution. Our method is closely related to normalizing flow and diffusion probabilistic models and can be viewed as a combination of the two. Compared with normalizing flow, diffusion normalizing flow is able to learn distributions with sharp boundaries. Compared with diffusion probabilistic models, diffusion normalizing flow requires fewer discretization steps and thus has better sampling efficiency. Our algorithm demonstrates competitive performance in both high-dimension data density estimation and image generation tasks.

## 1 Introduction

Generative model is a class of machine learning models used to estimate data distributions and sometimes generate new samples from the distributions [8, 35, 16, 37, 7]. Many generative models learn the data distributions by transforming a latent variable $\mathbf{z}$ with a tractable prior distribution to the data space [8, 35, 32]. To generate new samples, one can sample from the latent space and then follow the transformation to the data space. There exist a large class of generative models where the latent space and the data space are of the same dimension. The latent variable and the data are coupled through trajectories in the same space. These trajectories serve two purposes: in the forward direction $\mathbf{x} \rightarrow \mathbf{z}$, the trajectories infer the posterior distribution in the latent space associated with a given data sample $\mathbf{x}$, and in the backward direction $\mathbf{z} \rightarrow \mathbf{x}$, it generates new samples by simulating the trajectories starting from the latent space. This type of generative model can be roughly divided into two categories, depending on whether these trajectories connecting the latent space and the data space are deterministic or stochastic.

When deterministic trajectories are used, these generative models are known as flow-based models. The latent space and the data space are connected through an invertible map, which could either be realized by the composition of multiple invertible maps [35, 8, 20] or a differential equation [4, 14]. In these models, the probability density at each data point can be evaluated explicitly using the change of variable theorem, and thus the training can be carried out by minimizing the negative log-likelihood (NLL) directly. One limitation of the flow-based model is that the invertible map parameterized by neural networks used in it imposes topological constraints on the transformation from $\mathbf{z}$ to $\mathbf{x}$. Such limitation affects the performance significantly when the prior distribution on $\mathbf{z}$ is a simple unimodal distribution such as Gaussian while the target data distribution is a well-separated multi-modal distribution, i.e., its support has multiple isolated components. In [6], it is shown that there are some fundamental issues of using well-conditioned invertible functions to approximate such complicated multi-modal data distributions.

35th Conference on Neural Information Processing Systems (NeurIPS 2021).

When stochastic trajectories are used, the generative models are often known as the diffusion model [38]. In a diffusion model, a prespecified stochastic forward process gradually adds noise into the data to transform the data samples into simple random variables. A separate backward process is trained to revert this process to gradually remove the noise from the data to recover the original data distributions. When the forward process is modeled by a stochastic differential equation (SDE), the optimal backward SDE [1] can be retrieved by learning the score function [39, 40, 17, 2]. When the noise is added to the data sufficiently slow in the forward process, the backward diffusion can often revert the forward one reasonably well and is able to generate high fidelity samples. However, this also means that the trajectories have to be sufficiently long with a large number of time-discretization steps, which leads to slow training and sampling. In addition, since the forward process is fixed, the way noise is added is independent of the data distribution. As a consequence, the learned model may miss some complex but important details in the data distribution, as we will explain later.

In this work, we present a new generative modeling algorithm that resembles both the flow-based models and the diffusion models. It extends the normalizing flow method by gradually adding noise to the sampling trajectories to make them stochastic. It extends the diffusion model by making the forward process from $\mathbf{x}$ to $\mathbf{z}$ trainable. Our algorithm is thus termed *Diffusion Normalizing Flow (DiffFlow)*. The comparisons and relations among DiffFlow, normalizing flow, and diffusion models are shown in Figure 1. When the noise in DiffFlow shrinks to zero, DiffFlow reduces to a standard normalizing flow. When the forward process is fixed to some specific type of diffusion, DiffFlow reduces to a diffusion model.

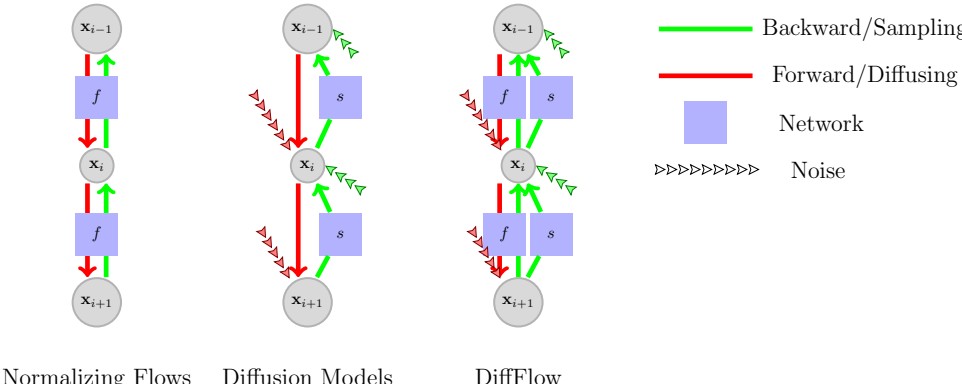

Figure 1: The schematic diagram for normalizing flows, diffusion models, and DiffFlow. In normalizing flow, both the forward and the backward processes are deterministic. They are the inverse of each other and thus collapse into a single process. The diffusion model has a fixed forward process and trainable backward process, both are stochastic. In DiffFlow, both the forward and the backward processes are trainable and stochastic.

In DiffFlow, the forward and backward diffusion processes are trained simultaneously by minimizing the distance between the forward and the backward process in terms of the Kullback-Leibler (KL) divergence of the induced probability measures [42]. This cost turns out to be equivalent to (see Section 3 for a derivation) the (amortized) negative evidence lower bound (ELBO) widely used in variational inference [21]. One advantage to use the KL divergence directly is that it can be estimated with no bias using sampled trajectories of the diffusion processes. The KL divergence in the trajectory space also bounds the KL divergence of the marginals, providing an alternative method to bound the likelihood (see Section 3 for details). To summarize, we have made the following contributions.
1. We propose a novel density estimation model termed diffusion normalizing flow (DiffFlow) that extends both the flow-based models and the diffusion models. The added stochasticity in DiffFlow boosts the expressive power of the normalizing flow and results in better performance in terms of sampling quality and likelihood. Compared with diffusion models, DiffFlow is able to learn a forward diffusion process to add noise to the data adaptively and more efficiently. This avoids adding noise to regions where noise is not so desirable. The learnable forward process also shortens the trajectory length, making the sampling much faster than standard diffusion models (We observe a 20 times speedup over diffusion models without decreasing sampling quality much).
2. We develop a stochastic adjoint algorithm to train the DiffFlow model. This algorithm evaluates the objective function and its gradient sequentially along the trajectory. It avoids storing all the

intermediate values in the computational graph, making it possible to train DiffFlow for high-dimensional problems.

3. We apply the DiffFlow model to several generative modeling tasks with both synthetic and real datasets, and verify the performance of DiffFlow and its advantages over other methods.

## 2 Background

Below we provide a brief introduction to normalizing flows and diffusion models. In both of these models, we use $\tau = \{\mathbf{x}(t), 0 \leq t \leq T\}$ to denote trajectories from the data space to the latent space in the continuous-time setting, and $\tau = \{\mathbf{x}_0, \mathbf{x}_1, \cdots, \mathbf{x}_N\}$ in the discrete-time setting.

**Normalizing Flows:** The trajectory in normalizing flows is modeled by a differential equation

$$\dot{\mathbf{x}} = \mathbf{f}(\mathbf{x}, t, \theta), \tag{1}$$

parameterized by $\theta$. This differential equation starts from random $\mathbf{x}(0) = \mathbf{x}$ and ends at $\mathbf{x}(T) = \mathbf{z}$. Denote by $p(\mathbf{x}(t))$ the probability distribution of $\mathbf{x}(t)$, then under mild assumptions, it evolves following [4]

$$\frac{\partial \log p(\mathbf{x}(t))}{\partial t} = -\mathrm{tr}(\frac{\partial \mathbf{f}}{\partial \mathbf{x}}). \tag{2}$$

Using this relation (1) (2) one can compute the likelihood of the model at any data point $\mathbf{x}$.

In the discrete-time setting, the map from $\mathbf{x}$ to $\mathbf{z}$ in normalizing flows is a composition of a collection of bijective functions as $F = F_N \circ F_{N-1} \cdots F_2 \circ F_1$. The trajectory $\tau = \{\mathbf{x}_0, \mathbf{x}_1, \cdots, \mathbf{x}_N\}$ satisfies

$$\mathbf{x}_i = F_i(\mathbf{x}_{i-1}, \theta), \quad \mathbf{x}_{i-1} = F_i^{-1}(\mathbf{x}_i, \theta) \tag{3}$$

for all $i$. Similar to Equation (2), based on the rule for change of variable, the log-likelihood of any data samples $\mathbf{x}_0 = \mathbf{x}$ can be evaluated as

$$\log p(\mathbf{x}_0) = \log p(\mathbf{x}_N) - \sum_{i=1}^{N} \log |\det(\frac{\partial F_i^{-1}(\mathbf{x}_i)}{\partial \mathbf{x}_i})|. \tag{4}$$

Since the exact likelihood is accessible in normalizing flows, these models can be trained by minimizing the negative log-likelihood directly.

**Diffusion Models:** The trajectories in diffusion models are modeled by stochastic differential equations. More explicitly, the forward process is of the form

$$d\mathbf{x} = \mathbf{f}(\mathbf{x}, t)dt + g(t)d\mathbf{w}, \tag{5}$$

where the *drift* term $\mathbf{f} : \mathbb{R}^d \times \mathbb{R} \to \mathbb{R}^d$ is a vector-valued function, and the *diffusion* coefficient $g : \mathbb{R} \to \mathbb{R}$ is a scalar function (in fact, $g$ is often chosen to be a constant). Here $\mathbf{w}$ denotes the standard Brownian motion. The forward process is normally a simple linear diffusion process [38, 16]. The forward trajectory $\tau$ can be sampled using (5) initialized with the data distribution. Denote by $p_F$ the resulting probability distribution over the trajectories. With a slight abuse of notation, we also use $p_F$ to denote the marginal distribution of the forward process.

The backward diffusion from $\mathbf{z} = \mathbf{x}(T)$ to $\mathbf{x} = \mathbf{x}(0)$ is of the form

$$d\mathbf{x} = [\mathbf{f}(\mathbf{x}, t) - g^2(t)\mathbf{s}(\mathbf{x}, t, \theta)]dt + g(t)d\mathbf{w}. \tag{6}$$

It is well-known that when $\mathbf{s}$ coincides with the score function $\nabla \log p_F$, and $\mathbf{x}(T)$ in the forward and backward processes share the same distribution, the distribution $p_B$ induced by the backward process (6) is equal to $p_F$ [1],[29, Chapter 13]. To train the score network $\mathbf{s}(\mathbf{x}, t, \theta)$, one can use the KL divergence between $p_F$ and $p_B$ as an objective function to reduce the difference between $p_F$ and $p_B$. When the difference is sufficiently small, $p_F$ and $p_B$ should have similar distribution over $\mathbf{x}(0)$, and one can then use the backward diffusion (6) to sample from the data distribution.

In the discrete setting, the trajectory distributions can be more explicitly written as

$$p_F(\tau) = p_F(\mathbf{x}_0) \prod_{i=1}^{N} p_F(\mathbf{x}_i | \mathbf{x}_{i-1}), \quad p_B(\tau) = p_B(\mathbf{x}_T) \prod_{i=1}^{N} p_B(\mathbf{x}_{i-1} | \mathbf{x}_i). \tag{7}$$

The KL divergence between $p_F$ and $p_B$ can be decomposed according to this expression (7). Most diffusion models use this decomposition, and meanwhile take advantage of the simple structure of the forward process (5), to evaluate the objective function in training [39, 40, 41].

## 3 Diffusion normalizing flow

We next present our diffusion normalizing flow models. Similar to diffusion models, the DiffFlow models also has a forward process

$$d\mathbf{x} = \mathbf{f}(\mathbf{x}, t, \theta)dt + g(t)d\mathbf{w}, \tag{8}$$

and a backward process

$$d\mathbf{x} = [\mathbf{f}(\mathbf{x}, t, \theta) - g^2(t)\mathbf{s}(\mathbf{x}, t, \theta)]dt + g(t)d\mathbf{w}. \tag{9}$$

The major difference is that, instead of being a fixed linear function as in most diffusion models, the drift term $\mathbf{f}$ is also learnable in DiffFlow. The forward process is initialized with the data samples at $t = 0$ and the backward process is initialized with a given noise distribution at $t = T$. Our goal is to ensure the distribution of the backward process at time $t = 0$ is close to the real data distribution. That is, we would like the difference between $p_B(\mathbf{x}(0))$ and $p_F(\mathbf{x}(0))$ to be small.

To this end, we use the KL divergence between $p_B(\tau)$ and $p_F(\tau)$ over the trajectory space as the training objective function. Since

$$KL(p_F(\mathbf{x}(t))|p_B(\mathbf{x}(t))) \le KL(p_F(\tau)|p_B(\tau)) \tag{10}$$

for any $0 \le t \le T$ by data processing inequality, small difference between $p_B(\tau)$ and $p_F(\tau)$ implies small difference between $p_B(\mathbf{x}(0))$ and $p_F(\mathbf{x}(0))$ in terms of KL divergence (more details are included in Appendix B).

### 3.1 Implementation

In real implementation, we discretize the forward process (8) and the backward process (9) as

$$\mathbf{x}_{i+1} = \mathbf{x}_i + \mathbf{f}_i(\mathbf{x}_i)\Delta t_i + g_i \delta_i^F \sqrt{\Delta t_i} \tag{11}$$

$$\mathbf{x}_i = \mathbf{x}_{i+1} - [\mathbf{f}_{i+1}(\mathbf{x}_{i+1}) - g_{i+1}^2 \mathbf{s}_{i+1}(\mathbf{x}_{i+1})]\Delta t_i + g_{i+1}\delta_i^B \sqrt{\Delta t_i}, \tag{12}$$

where $\delta_i^F, \delta_i^B \sim \mathcal{N}(0, \mathbf{I})$ are unit Gaussian noise, $\{t_i\}_{i=0}^N$ are the discretization time points, and $\Delta t_i = t_{i+1} - t_i$ is the step size at the $i$-th step. Here we have dropped the dependence on the parameter $\theta$ to simplify the notation. With this discretization, the KL divergence between trajectory distributions becomes

$$KL(p_F(\tau)|p_B(\tau)) = \underbrace{\mathbb{E}_{\tau \sim p_F}[\log p_F(\mathbf{x}_0)]}_{L_0} + \underbrace{\mathbb{E}_{\tau \sim p_F}[-\log p_B(\mathbf{x}_N)]}_{L_N} + \sum_{i=1}^{N-1} \underbrace{\mathbb{E}_{\tau \sim p_F}[\log \frac{p_F(\mathbf{x}_i|\mathbf{x}_{i-1})}{p_B(\mathbf{x}_{i-1}|\mathbf{x}_i)}]}_{L_i}. \tag{13}$$

The term $L_0$ in (13) is a constant determined by entropy of the dataset as

$$\mathbb{E}_{\tau \sim p_F}[\log p_F(\mathbf{x}_0)] = \mathbb{E}_{\mathbf{x}_0 \sim p_F}[\log p_F(\mathbf{x}_0)] =: -H(p_F(x(0))).$$

The term $L_N$ is easy to calculate since $p_B(x_N)$ is a simple distribution, typically standard Gaussian distribution.

To evaluate $L_{1:N-1}$, we estimate it over sampled trajectory from the forward process $p_F$. For a given trajectory $\tau$ sampled from $p_F(\tau)$, we need to calculate $p_B(\tau)$ along the same trajectory. To this end, a specific group of $\{\delta_i^B\}$ is chosen such that the same trajectory can be reconstructed from the backward process. Thus, $\delta_i^B$ satisfies

$$\delta_i^B(\tau) = \frac{1}{g_{i+1}\sqrt{\Delta t}} \left[ \mathbf{x}_i - \mathbf{x}_{i+1} + [\mathbf{f}_{i+1}(\mathbf{x}_{i+1}) - g_{i+1}^2 \mathbf{s}_{i+1}(\mathbf{x}_{i+1})]\Delta t \right]. \tag{14}$$

Since $\delta_i^B$ is a Gaussian noise, the negative log-likelihood term $p_B(\mathbf{x}_i|\mathbf{x}_{i+1})$ is equal to $\frac{1}{2}(\delta_i^B(\tau))^2$ after dropping constants (see more details in Appendix B). In view of the fact that the expectation of $\sum_i \frac{1}{2}(\delta_i^F(\tau))^2$ remains a constant, minimizing Equation (13) is equivalent to minimizing the following loss (see Appendix C for the full derivation):

$$L := \mathbb{E}_{\tau \sim p_F}[-\log p_B(\mathbf{x}_N) + \sum_i \frac{1}{2}(\delta_i^B(\tau))^2] = \mathbb{E}_{\delta^F; \mathbf{x}_0 \sim p_0}[-\log p_B(\mathbf{x}_N) + \sum_i \frac{1}{2}(\delta_i^B(\tau))^2], \tag{15}$$

where the last equality is based on a reparameterization trick [21]. We can minimize the loss in Equation (15) with Monto Carlo gradient estimation as in Algorithm 1.

---

**Algorithm 1** Training

---

**repeat**
    $\mathbf{x}_0 \sim$ Real data distribution
    $\delta_{1:N}^F \sim \mathcal{N}(0, \mathbf{I})$
    Discrete timestamps: $t_{i=0}^N$
    Sample: $\tau = \{\mathbf{x}_i\}_{i=0}^N$ based on $\delta_{1:N}^F$
    Gradient descent step $\nabla_\theta[-\log p_B(\mathbf{x}_N) + \sum_i \frac{1}{2}(\delta_i^B(\tau))^2]$
**until** converged

---

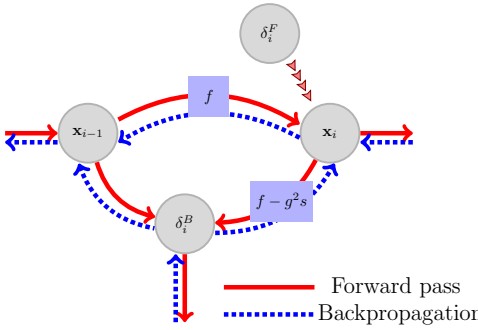

---

**Algorithm 2** Stochastic Adjoint Algorithm for DiffFlow

---

1: **Input:** Forward trajectory $\{\mathbf{x}_i\}_{i=0}^N$
2: $\frac{\partial L}{\partial \mathbf{x}_N} = \frac{1}{2}\frac{\partial(\delta_N^B(\tau))^2}{\partial \mathbf{x}_N} - \frac{\partial \log p_B(\mathbf{x}_N)}{\mathbf{x}_N}$
3: $\frac{\partial L}{\partial \theta} = 0$
4: **for** $i = N, N-1, \cdots, 1$ **do**
5: $\quad \frac{\partial L}{\partial \mathbf{x}_{i-1}} = \left(\frac{\partial L}{\partial \mathbf{x}_i} + \frac{1}{2}\frac{\partial(\delta_i^B(\tau))^2}{\partial \mathbf{x}_i}\right)\frac{\partial \mathbf{x}_i}{\partial \mathbf{x}_{i-1}} + \frac{1}{2}\frac{\partial(\delta_i^B(\tau))^2}{\partial \mathbf{x}_{i-1}}$
6: $\quad \frac{\partial L}{\partial \theta} += \frac{1}{2}\frac{\partial(\delta_i^B(\tau))^2}{\partial \theta} + \left(\frac{\partial L}{\partial \mathbf{x}_i} + \frac{1}{2}\frac{\partial(\delta_i^B(\tau))^2}{\partial \mathbf{x}_i}\right)\frac{\partial \mathbf{x}_i}{\partial \theta}$
7: **end for**

---

Figure 2: Gradient Flowchart.

### 3.2 Stochastic Adjoint method

One challenge in training DiffFlow is memory consumption. When a naive backpropagation strategy is used, the memory consumption explodes quickly. Indeed, differentiating through the operations of the forward pass requires unrolling networks $N$ times and caching all network intermediate values for every step, which prevents this naive implementation of DiffFlow from being applied in high dimensional applications. Inspired by the adjoint method in Neural ODE [4], we propose the adjoint variable $\frac{\partial L}{\partial \mathbf{x}_i}$ and a stochastic adjoint algorithm that allows training the DiffFlow model with reduced memory consumption. In this adjoint method, we cache intermediate states $\mathbf{x}_i$ and, based on these intermediate states, reproduce the whole process, including $\delta_i^F, \delta_i^B$ as well as $f_i, s_i$ exactly.

We note another similar approach [26] of training SDEs caches random noise $d\mathbf{w}$ and further takes advantage of the pseudo-random generator to save memory for the intermediate noises, resulting in constant memory consumption However, the approach can not reproduce exact trajectories due to time discretization error and requires extra computation to recover $d\mathbf{w}$ from the pseudo-random generator. We found that in our image experiments in Section 4, the cached $\{\mathbf{x}_i\}$ consumes only about 2% memory out of all the memory being used during training. Due to the accuracy and acceptable memory overhead, the introduced stochastic adjoint approach is a better choice for DiffFlow. We summarize the method in Algorithm 2 and Figure 2. We also include the PyTorch [34] implementation in the supplemental material.

### 3.3 Time discretization and progressive training

We propose two time discretization schemes for training DiffFlow: fixed timestamps $L_\beta$ and flexible timestamps $\hat{L}_\beta$. For fixed timestamps, the time interval $[0, T]$ is discretized with fixed schedule $t_i = (\frac{i}{N})^\beta T$. With such fixed time discretization over batches, we denote the loss function as $L_\beta$. Empirically, we found $\beta = 0.9$ works well. This choice of $\beta$ increases stepsize $\Delta t_i$ when the forward process approaches $\mathbf{z} = \mathbf{x}_N$ and provides higher resolution when the backward process is close to $\mathbf{x}_0$. We found such discretization generates samples with good quality and high fidelity details. The choice of polynomial function is arbitrary; other functions of similar sharps may work as well.

Figure 3: $\Delta t_i$ of $L_\beta$ and $\hat{L}_\beta$

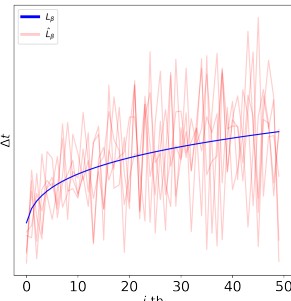

In the flexiable timestamps scheme, we train different batches with different time discretization points. Specifically, $t_i$ is sampled uniformly from the interval $[(\frac{i-1}{N-1})^\beta T, (\frac{i}{N-1})^\beta T]$ for each batch. We denote the training objective function with such random time discretization as $\hat{L}_\beta$. We empirically found such implementation results in lower loss and better stability when we conduct progressive training where we increase $N$ gradually as training progresses. In progressive training, we refine the forward and backward processes as $N$ increase. This training scheme can significantly save training time compared with the other method that uses a fixed large $N$ during the whole process. Empirically, we found that progressive training can speed up the training up to 16 times.

To understand why such random time discretization scheme is more stable, we hypothesis that this choice encourages a smoother $\mathbf{f}, \mathbf{s}$ since it seeks functions $\mathbf{f}, \mathbf{s}$ to reduce objective loss under different sets of $\{t_i\}$ instead of a specific $\{t_i\}$. We illustrate fixed timestamps in $L_\beta$ and a realization of random discretization in $\hat{L}_\beta$ in Figure 3 with $\beta = 0.9$.

## 3.4 Learnable forward process

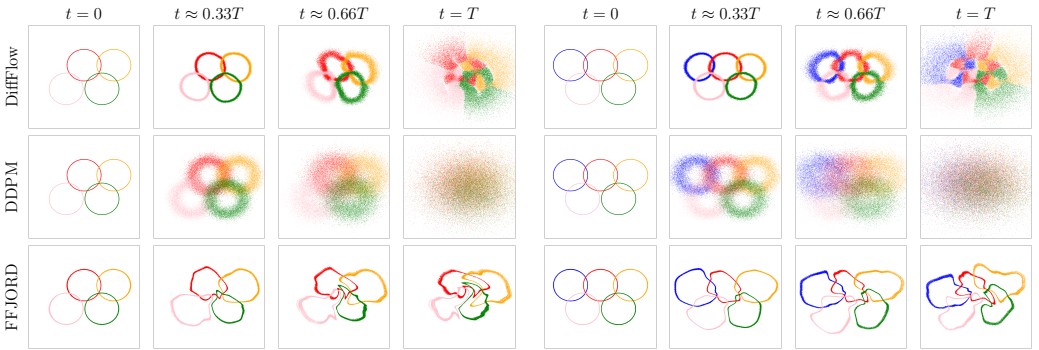

Figure 4: Illustration of forwarding trajectories of DiffFlow, DDPM, and FFJORD. Each row shows two trajectories of transforming data distributions, four rings, and Olympics rings, to a base distribution. Different modes of densities are in different colors. Though FFJORD adjusts forward process based on data, its bijective property prevents the approach from expanding density support to the whole space. DDPM can transform data distributions into Gaussian distribution but a data-invariant way of adding noise can corrupt the details of densities, e.g., the densities at the intersections of the rings. DiffFlow not only transforms data into base distribution but also keeps the topological information of the original datasets. Points from the same ring are transformed into continental plates instead of being distributed randomly.

The forward process not only is responsible for driving data into latent space, but also provides enough supervised information to learning backward process. Thanks to bijective property, NFs can reconstruct data exactly but there is no guarantee that it can reach the standard Gaussian. At the other extreme, Denoising diffusion probabilistic models (DDPM) [17] adopt a data-invariant forward diffusing schema, ensuring that $\mathbf{x}_N$ is Gaussian. DDPM can even reach Gaussian in one step with $N = 1$, which output noise disregarding data samples. However, backward process will be difficult to learn if data is destroyed in one step. Therefore, DDPM adds noise slowly and often needs more than one thousand steps for diffusion.

The forward module of DiffFlow is a combination of normalizing flow and diffusion models. We show the comparision in fitting toy 2D datasets in Figure 4. We are especially interested in data with well-separated modes and sharp density boundaries. Those properties are believed to appear in various datasets. As stated by manifold hypothesis [36], real-world data lie on low-dimensional manifold [28] embedded in a high-dimensional space. To construct the distributions in Figure 4, we rotate the 1-d Gaussian distribution $\mathcal{N}(1, 0.001^2)$ around the center to form a ring and copy the rings to different locations.

As a bijective model, FFJORD [14] struggles to diffuse the concentrated density mass into a Gaussian distribution. DiffFlow overcomes expressivity limitations of the bijective constraint by adding noise. As added noise shrinks to zero, the DiffFlow has no stochasticity and degrades to a flow-based model. Based on this fact, we present the following theorem with proof in Appendix A.

**Theorem 1.** *As diffusion coefficients $g_i \to 0$, DiffFlow reduces to Normalizing Flow. Moreover, minimizing the objective function in Equation (13) is equivalent to minimizing the negative log-likelihood as in Equation (4).*

DDPM [17] uses a fixed noising transformation. Due to the data-invariant approach $p(\mathbf{x}_T|\mathbf{x}_0) = p(\mathbf{x}_T)$, points are diffused in the same way even though they may appear in different modes or different datasets. We observe that sharp details are destroyed quickly in DDPM diffusion, such as the intersection regions between rings. However, with the help of learnable transformation, DiffFlow diffuses in a much efficient way. The data-dependent approach shows different diffusion strategies on different modes and different datasets. Meanwhile, similar to NFs, it keeps some topological information for learning backward processes. We include more details about the toy sample in Section 4.

## 4 Experiments

We evaluate the performance of DiffFlow in sample quality and likelihood on test data. To evaluate the likelihood, we adopt the marginals distribution equivalent SDEs

$$dx = [\mathbf{f}(\mathbf{x}, t, \theta) - \frac{1 + \lambda^2}{2} g^2(t)\mathbf{s}(\mathbf{x}, t, \theta)]dt + \lambda g(t)d\mathbf{w}, \qquad (16)$$

with $\lambda \geq 0$ (Proof see Appendix H). When $\lambda = 0$, it reduces to probability ODE [41]. The ODE provides an efficient way to evaluate the density and negative log-likelihood. For any $0 \leq \lambda \leq 1$, the above SDE can be used for sampling. Empirically, we found $\lambda = 1$ has the best performance.

### 4.1 Synthetic 2D examples

We compare the performance of DiffFlow and existing diffusion models and NFs on estimating the density of 2-dimensional data. We compare the forward trajectories of DiffFlow, DDPM [17] and FFJORD [14][1] in Figure 4 and its sampling performance in Figure 5. To make a fair comparison, we build models with comparable network sizes, around 90k learnable parameters. We include more training and model details in Appendix E.

All three algorithms have good performance on datasets whose underlying distribution has smooth density, such as 2 spirals. However, when we shrink the support of samples or add complex patterns, performance varies significantly. We observe that FFJORD leaks many samples out of the main modes and datasets with complex details and sharp density exacerbates the disadvantage.

DDPM has higher sample quality but blurs density details, such as intersections between rings, areas around leaves of the Fractal tree, and boxes in Sierpiński Carpet. The performance is within expectation given that details are easy to be destroyed and ignored with the data-invariant noising schema. On the less sharp dataset, such as 2 Spirals and Checkerboard, its samples align with data almost perfectly.

DiffFlow produces the best samples (according to a human observer). We owe the performance to the flexible noising forward process. As illustrated in Figure 4, DiffFlow provides more clues and retains detailed patterns longer for learning its reverse process. We also report a comprehensive comparison of the negative likelihood and more analysis in Appendix E. DiffFlow has a much lower negative likelihood, especially on sharp datasets.

### 4.2 Density estimation on real data

We perform density estimation experiments on five tabular datasets [33]. We employ the probability flow to evaluate the negative log-likelihood. We find that our algorithm has better performance in most datasets than most approaches trained by directly minimizing negative log-likelihood, including NFs and autoregressive models. DiffFlow outperforms FFJORD by a wide margin on all datasets except HEPMASS. Compared with autoregressive models, it excels NAF [18] on all but GAS. Those models require $\mathcal{O}(d)$ computations to sample from. Meanwhile, DiffFlow is quite effective in achieving such performance with MLPs that have less than 5 layers. We include more details in Appendix F

### 4.3 Image generation

In this section, we report the quantitative comparison and qualitative performance of our method and existing methods on common image datasets, MNIST [24] and CIFAR-10 [23]. We use the same

---

[1]Implementation of FFJORD and DDPM are based on official codebases.

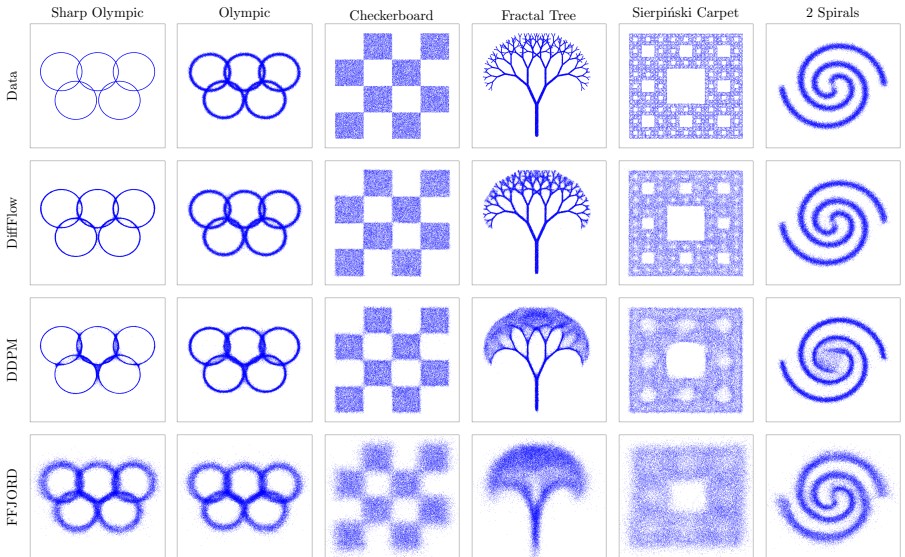

Figure 5: Samples from DiffFlow, DDPM and FFJORD on 2-D datasets. All three models have reasonable performance on datasets that have smooth underlying distributions. But only DiffFlow is capable to capture complex patterns and provides sharp samples when dealing with more challenging datasets.

| Dataset | POWER | GAS | HEPMASS | MINIBOONE | BSDS300 |
|---|---|---|---|---|---|
| RealNVP [8] | -0.17 | -8.33 | 18.71 | 13.55 | -153.28 |
| FFJORD [14] | -0.46 | -8.59 | **14.92** | 10.43 | -157.40 |
| DiffFlow (ODE) | **-1.04** | -10.45 | 15.04 | **8.06** | **-157.80** |
| MADE [11] | 3.08 | -3.56 | 20.98 | 15.59 | -148.85 |
| MAF [33] | -0.24 | -10.08 | 17.70 | 11.75 | -155.69 |
| TAN [31] | -0.48 | -11.19 | 15.12 | 11.01 | -157.03 |
| NAF [18] | -0.62 | **-11.96** | 15.09 | 8.86 | -157.73 |

Table 1: Average negative log-likelihood (in nats) on tabular datasets [33] for density estimation (lower is better).

unconstrained U-net style model as used successfully by Ho et al. [17] for drift and score network. We reduce the network size to half of the original DDPM network so that the total number of trainable parameters of DiffFlow and DDPM are comparable. We use small $N = 10$ at the beginning of training and slowly increase to large $N$ as training proceeds. The schedule of $N$ reduces the training time greatly compared with using large $N$ all the time. We use constants $g_i = 1$ and $T = 0.05$ for MNIST and CIFAR10, and $N = 30$ for sampling MNIST data and $N = 100$ for sampling CIFAR10. As it is reported by Jolicoeur-Martineau et al. [19], adding noise at the last step will significantly lower sampling quality, we use one single denoising step at the end of sampling with Tweedie's formula [10].

We report negative log-likelihood (NLL) in bits per dimension or negative ELBO if NLL is unavailable. On MNIST, we achieve competitive performance on NLL as in Table 2. We show the uncurated samples from DiffFlow in Figure 6 and Figure 7. On CIFAR-10, DiffFlow also achieves competitive NLL performance as shown in Table 3. DiffFlow performs better than normalizing flows and DDPM models, but is slightly worse than DDPM++(sub, deep, sub-vp) and Improved DDPM. However, these approaches conduct multiple architectural improvements and use much deeper and wider networks. We also report the popular sample metric, Fenchel Inception Distance (FID) [15]. DiffFLow has a lower FID score than normalizing flows and has competitive performance compared with DDPM trained with unweighted variational bounds, DDPM and Improved DDPM. It is worse than DDPM trained with reweighted loss, DDPM ($L_s$), DDPM cont, and DDPM++ [17, 41, 30]. Besides, sampling

Table 2: NLL on MNIST

| Model | NLL ($\downarrow$) |
| --- | --- |
| RealNVP [8] | 1.06 |
| Glow [20] | 1.05 |
| FFJORD [14] | 0.99 |
| ResFlow [5] | 0.97 |
| DiffFlow | 0.93 |

Figure 6: MNIST Samples

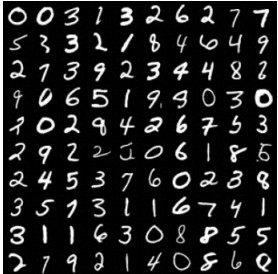

Figure 7: CIFAR10 Samples

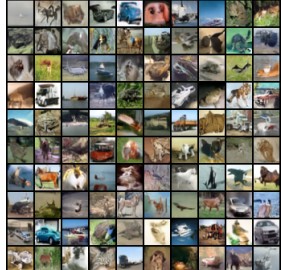

quality with different sampling steps $N$ are compared in Table 4 [2]. The advantage of DiffFlow is clear when we compare relative FIDs degeneracy ratio with $N = 100$ respectively. DiffFlow is able to retain better sampling quality when decreasing $N$. Full details on architectures used, training setup details, and more samples can be found in Appendix G.

Table 3: NLLs and FIDs on CIFAR-10.

| Model | NLL($\downarrow$) | FID ($\downarrow$) |
| --- | --- | --- |
| RealNVP [8] | 3.49 | - |
| Glow [20] | 3.35 | 46.90 |
| Flow++ [16] | 3.29 | - |
| FFJORD [14] | 3.40 | - |
| ResFlow [5] | 3.28 | 46.37 |
| DDPM ($L$) [17] | $\leq 3.70$ | 13.51 |
| DDPM ($L_s$) [17] | $\leq 3.75$ | 3.17 |
| DDPM ($L_s$)(ODE) [41] | 3.28 | 3.37 |
| DDPM cont. (sub-VP) [41] | 3.05 | 3.56 |
| DDPM++ (sub-VP) [41] | 3.02 | 3.16 |
| DDPM++ (deep, sub-VP) [41] | 2.99 | 2.94 |
| Improved DDPM [30] | $\leq 2.94$ | 11.47 |
| DiffFlow ($L_\beta$) | $\leq 3.71$ | 13.87 |
| DiffFlow ($\hat{L}_\beta$) | $\leq 3.67$ | 13.43 |
| DiffFlow ($\hat{L}_\beta$, ODE) | 3.04 | 14.14 |

Table 4: FIDs with various $N$

| $N$ | DiffFlow | DDPM ($L$) | DDPM ($L_s$) | DDIM |
| --- | --- | --- | --- | --- |
| 5 | 28.31 | 373.51 | 370.23 | 44.69 |
| 10 | 22.56 | 364.64 | 365.12 | 18.62 |
| 20 | 17.98 | 138.84 | 135.44 | 10.89 |
| 50 | 14.72 | 47.12 | 34.56 | 7.01 |
| 100 | 13.43 | 22.23 | 10.04 | 5.63 |

Table 5: Relative FIDs degeneracy ratio

| $N$ | DiffFlow | DDPM ($L$) | DDPM ($L_s$) | DDIM |
| --- | --- | --- | --- | --- |
| 5 | 2.12 | 16.80 | 37.02 | 7.94 |
| 10 | 1.68 | 16.40 | 36.12 | 3.31 |
| 20 | 1.34 | 6.24 | 13.54 | 1.93 |
| 50 | 1.10 | 2.12 | 3.45 | 1.24 |
| 100 | 1.0 | 1.0 | 1.0 | 1.0 |

## 5 Related work

Normalizing flows [8, 35] have recently received lots of attention due to its exact density evaluation and ability to model high dimensional data [20, 9]. However, the bijective requirement poses limitations on modeling complex data, both empirically and theoretically [42, 6]. Some works attempt to relax the bijective requirement; discretely index flows [9] use domain partitioning with only locally invertible functions. Continuously indexed flows [6] extend discretely indexing to a continuously indexing approach. As pointed out in Stochastic Normalizing Flows (SNF) [42], stochasticity can effectively improve the expressive power of the flow-based model in low dimension applications. The architecture used in SNF, which requires known underlying energy models, presents challenges for density learning tasks; SNF is designed for sampling from unnormalized probability distribution instead of density estimation. Besides, even with ideal networks and infinite amount of data, due to the predefined stochstic block being used, SNF cannot find models with aligned forward and backward distribution as DiffFlow.

When it comes to stochastic trajectories, minimizing the distance between trajectory distributions has been explored in existing works. Denoising diffusion model [38] uses a fixed linear forward diffusion schema and reparameterizes the KL divergence such that minimizing loss is possible without computing whole trajectories. Diffusion models essentially corrupt real data iteratively and learn to remove the noise when sampling. Recently, Diffusion models have shown the capability to model

---

[2]The performance of DDPM is evaluated based on the officially released checkpoint with $L_s$ denotes for $L_{simple}$ in the original paper.

high-dimensional data distribution, such as images [17, 40], shapes [3], text-to-speech [22]. Lately, the Score-based model [41] provides a unified framework for score-matching methods and diffusion models based on stochastic calculus. The diffusion processes and sampling processes can be viewed as forwarding SDE and reverse-time SDE. Thanks to the linear forward SDE being used in DDPM, the forward marginal distributions have a closed-form and are suitable for training score functions on large-scale datasets. Also, due to the reliance on fixed linear forward process, it takes thousands of steps to diffuse data and generate samples. DiffFlow considers general SDEs and nosing and sampling are more efficient.

Existing Neural SDE approaches suffer from poor scaling properties. Backpropagating through solver [12] has a linear memory complexity with the number of steps. The pathwise approach [13] scales poorly in computation complexity. Our stochastic adjoint approach shares a similar spirit with SDE adjoint sensitivity [25]. The choice of caching noise requires high resolution of time discretization and prevents the approach from scaling to high dimension applications. By caching the trajectory states, DiffFlow can use a coarser discretization and deploy on larger dimension problems and problems with more challenging densities. The additional memory footprint is negligible compared with the other network memory consumption in DiffFlow.

## 6 Limitations

While DiffFlow gains more flexibility due to the introduction of a learnable forward process, it loses the analytical form for $p_F(\mathbf{x}_t|\mathbf{x}_0)$ and thus the training less efficient compared with score-based loss [41]. Training DiffFlow relies on backpropagation through trajectories and is thus significantly slower than diffusion models with affine drift. Empirically, we found DiffFlow is about 6 times slower than DDPM in 2d toy examples, 55 times in MNIST, and 160 times in CIFAR10 without progressive training in Section 3.3. Though the stochastic adjoint method and progressive training help save memory footprint and reduce training time, the training of DiffFlow is still more expensive than DDPM and its variants. On the other hand, compared with normalizing flows, the extra noise in DiffFlow boosts the expressive power of the model with little extra cost. Though DiffFlow trained based on SDE, its marginal distribution equivalent ODE 2 shows much better performance than its counterpart trained with ODE [14]. It is interesting to investigate, both empirically and theoretically, the benefit in terms of expressiveness improvement caused by stochastic noise for training normalizing flows.

## 7 Conclusions

We proposed a novel algorithm, the diffusion normalizing flow (DiffFlow), for generative modeling and density estimation. The proposed method extends both the normalizing flow models and the diffusion models. Our DiffFlow algorithm has two trainable diffusion processes modeled by neural SDEs, one forward and one backward. These two SDEs are trained jointly by minimizing the KL divergence between them. Compared with most normalizing flow models, the added noise in DiffFlow relaxes the bijectivity condition in deterministic flow-based models and improves their expressive power. Compared with diffusion models, DiffFlow learns a more flexible forward diffusion that is able to transform data into noise more effectively and adaptively. In our experiments, we observed that DiffFlow is able to model distributions with complex details that are not captured by representative normalizing flow models and diffusion models, including FFJORD, DDPM. For CIFAR10 dataset, our DiffFlow method has worse performance than DDPM in terms of FID score. We believe our DiffFlow algorithm can be improved further by using different neural network architectures, different time discretizing method and different choices of time interval. We plan to explore these options in the near future.

Our algorithm is able to learn the distribution of high-dimensional data and then generate new samples from it. Like many other generative modeling algorithms, it may be potentially used to generate misleading data such as fake images or videos.

## Acknowledgements

The authors would like to thank the anonymous reviewers for useful comments. This work is supported in part by grants NSF CAREER ECCS-1942523 and NSF CCF-2008513.

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
