# A  Proof of Theorem 1: Normalizing Flow as a special case of DiffFlow

*Proof.* As $g(t) \to 0$, the stochastic trajectories become deterministic. The forward SDE in DiffFlow reduces to the ODE in Normalizing Flow. In the discrete implementation, DiffFlow reduces to a normalizing flow consisting of the discrete layers

$$F_i(\mathbf{x}, \theta) = \mathbf{x} + \mathbf{f}(\mathbf{x}, t_i, \theta)\Delta t_i \quad \mathbf{x}_i = F_i(\mathbf{x}_{i-1}, \theta). \tag{17}$$

In the following, we show that minimizing the KL divergence between trajectory distributions is equivalent to minimizing the negative log-likelihood in Normalizing flow as $g(t) \to 0^+$. The derivation is essentially the same as that in [42] except that our model is for density estimation instead of sampling [42].

The discrete forward dynamics of DiffFlow is $\mathbf{x}_i = F_i(\mathbf{x}_{i-1}) + g_i\delta_i^F\sqrt{\Delta t_i}$, and it is associated with the conditional distribution

$$p_F(\mathbf{x}_i|\mathbf{x}_{i-1}) = \mathcal{N}(\mathbf{x}_i - F_i(\mathbf{x}_{i-1}); \mathbf{0}, g_i^2\mathbf{I}\Delta t_i). \tag{18}$$

Denote the conditional distribution for the backward process by

$$p_B(\mathbf{x}_{i-1}|\mathbf{x}_i) = \frac{p_B(\mathbf{x}_{i-1})p_B(\mathbf{x}_i|\mathbf{x}_{i-1})}{\int p_B(\mathbf{x})p_B(\mathbf{x}_i|\mathbf{x})d\mathbf{x}}. \tag{19}$$

For a fixed forward process, minimizing the KL divergence implies that $p_B(\mathbf{x}_{i-1}|\mathbf{x}_i)$ is the posterior of $p_F(\mathbf{x}_i|\mathbf{x}_{i-1})$, that is, $p_F(\mathbf{x}_i|\mathbf{x}_{i-1}) = p_B(\mathbf{x}_i|\mathbf{x}_{i-1})$. This is possible when the diffusion intensity $g(t)$ is small. It follows that

$$\frac{p_F(\mathbf{x}_i|\mathbf{x}_{i-1})}{p_B(\mathbf{x}_{i-1}|\mathbf{x}_i)} = \frac{\int p_B(\mathbf{x})p_B(\mathbf{x}_i|\mathbf{x})d\mathbf{x}}{p_B(\mathbf{x}_{i-1})}. \tag{20}$$

In the deterministic limit $g_i \to 0^+$,

$$\lim_{g \to 0^+} \int p_B(\mathbf{x})p_B(\mathbf{x}_i|\mathbf{x}) = \lim_{g \to 0^+} \int p_B(\mathbf{x})\mathcal{N}(\mathbf{x}_i - F_i(\mathbf{x}); \mathbf{0}, g_i^2\mathbf{I}\Delta t_i)d\mathbf{x}$$

$$= \lim_{g \to 0^+} \int p_B(F_i^{-1}(\mathbf{x}'))\mathcal{N}(\mathbf{x}_i - \mathbf{x}'; \mathbf{0}, g_i^2\mathbf{I}\Delta t_i)|\det(\frac{\partial F_i^{-1}(\mathbf{x}')}{\partial \mathbf{x}'})|d\mathbf{x}'$$

$$= p_B(F_i^{-1}(\mathbf{x}_i))|\det(\frac{\partial F_i^{-1}(\mathbf{x}_i)}{\partial \mathbf{x}_i})|$$

$$= p_B(\mathbf{x}_{i-1})|\det(\frac{\partial F_i^{-1}(\mathbf{x}_i)}{\partial \mathbf{x}_i})|.$$

The second equality is based on the rule of change variable for the map $\mathbf{x}' = F_i^{-1}(\mathbf{x})$. Plugging the above and Equation (20) into Equation (13), we arrive at

$$\mathbb{E}_{\tau \sim p_F}[\log p_F(\mathbf{x}_0) - \log p_B(\mathbf{x}_N) + \sum_{i=1}^N \log|\det\frac{\partial F_i^{-1}(\mathbf{x}_i)}{\partial \mathbf{x}_i}|]. \tag{21}$$

Therefore minimizing the KL divergence between forward and backward trajectory distributions is equivalent to maximizing the log probability as in Equation (4). □

# B  Proofs for Eq (10) and Eq (14)

## B.1  Proof of Eq (10)

Eq (10) follows from the disintegration of measure theorem and the non-negativity of KL divergence. More specifically, by disintegration of measure,

$$p_F(\tau) = \int p_F(\tau|\mathbf{x}(t))p_F(\mathbf{x}(t))d\mathbf{x}(t) \quad p_B(\tau) = \int p_B(\tau|\mathbf{x}(t))p_B(\mathbf{x}(t))d\mathbf{x}(t).$$

Eq (10) follows that

$$KL(p_F(\tau)|p_B(\tau)) = KL(p_F(\mathbf{x}(t))|p_B(\mathbf{x}(t)) + \int KL(p_F(\tau|\mathbf{x}(t))|p_B(\tau|\mathbf{x}(t)))d\mathbf{x}(t) \geq KL(p_F(\mathbf{x}(t))|p_B(\mathbf{x}(t)).$$

## B.2  Proof for Eq (14) and log-likelihood of $p_B(\mathbf{x}_i|\mathbf{x}_{i+1})$

For a given trajectory $\tau = \{\mathbf{x}_i\}$ generated from forward dynamics, the density of such trajectory in the distribution induced by backward dynamics

$$p_B(\tau) = p_B(\mathbf{x}_T)\prod_{i=1}^N p_B(\mathbf{x}_{i-1}|\mathbf{x}_i).$$

And term $p_B(\mathbf{x}_i|\mathbf{x}_{i+1})$ can be written as

$$p_B(\mathbf{x}_i|\mathbf{x}_{i+1}) = \int p_B(\mathbf{x}_i|\mathbf{x}_{i+1}, \delta_i^B)\mathcal{N}(\delta_i^B)d\delta_i^B,$$

where $\mathcal{N}$ denotes the standard normal distribution. The distribution $p_B(\mathbf{x}_i|\mathbf{x}_{i+1})$ and $p_B(\mathbf{x}_i|\mathbf{x}_{i+1}, \delta_i^B)$ encode dynamics in Eq (9). Therefore density can be reformulated as

$$p_B(\mathbf{x}_i|\mathbf{x}_{i+1}, \delta_i^B) = \mathbb{I}_{\mathbf{x}_i = \mathbf{x}_{i+1} - [\mathbf{f}_{i+1}(\mathbf{x}_{i+1}) - g_{i+1}^2\mathbf{s}_{i+1}(\mathbf{x}_{i+1})]\Delta t_i + g_{i+1}\delta_i^B\sqrt{\Delta t_i}}$$

$$p_B(\mathbf{x}_i|\mathbf{x}_{i+1}) = \mathcal{N}\left(\frac{1}{g_{i+1}\sqrt{\Delta t}}\left[\mathbf{x}_i - \mathbf{x}_{i+1} + [\mathbf{f}_{i+1}(\mathbf{x}_{i+1}) - g_{i+1}^2\mathbf{s}_{i+1}(\mathbf{x}_{i+1})]\Delta t\right]\right)$$

Thus, the negative log-likelihood term $p_B(\mathbf{x}_i|\mathbf{x}_{i+1})$ is equal to $\frac{1}{2}(\delta_i^B(\tau))^2$ where $\delta_i^B(\tau)$ is given by Eq 14.

## C   Derivation of the loss function

The KL divergence between the distributions induced by the forward process and the backward process is

$$KL(p_F(\tau)|p_B(\tau)) = KL(p_F(\mathbf{x}_0)p_F(\tau|\mathbf{x}_0)|p_B(\mathbf{x}_N)p_B(\tau|\mathbf{x}_N)) \tag{22}$$

$$= \mathop{\mathbb{E}}_{p_F(\mathbf{x}_0)p_F(\tau|\mathbf{x}_0)}[\log p_F(\mathbf{x}_0) - \log p_B(\mathbf{x}_N) + \log\frac{p_F(\tau|\mathbf{x}_0)}{p_B(\tau|\mathbf{x}_N)}]$$

In discrete implementation, the density ratio between conditional forward and reverse process can be reformulated as

$$\frac{p_F(\tau|\mathbf{x}_0)}{p_B(\tau|\mathbf{x}_N)} = \frac{p_F(\mathbf{x}_N|\mathbf{x}_{N-1})\cdots p_F(\mathbf{x}_2|\mathbf{x}_1)p_F(\mathbf{x}_1|\mathbf{x}_0)}{p_B(\mathbf{x}_{N-1}|\mathbf{x}_N)\cdots p_B(\mathbf{x}_1|\mathbf{x}_2)p_B(\mathbf{x}_0|\mathbf{x}_1)}.$$

Since

$$\frac{p_F(\mathbf{x}_i|\mathbf{x}_{i-1})}{p_B(\mathbf{x}_{i-1}|\mathbf{x}_i)} = \frac{g_i\mathcal{N}(\delta_i^F|0, 1)}{g_{i+1}\mathcal{N}(\delta_i^B|0, 1)}.$$

and

$$\log\frac{p(\delta_i^F)}{p(\delta_i^B)} = \frac{1}{2}[(\delta_i^B)^2 - (\delta_i^F)^2],$$

we can rewrite the loss function as

$$KL(p_F(\tau)|p_B(\tau)) = -H(p_F(\mathbf{x}_0)) + \mathop{\mathbb{E}}_{\tau\sim p_F}[-\log p_B(\mathbf{x}_N) + \sum_{i=1}^{N}\log\frac{p_F(\mathbf{x}_i|\mathbf{x}_{i-1})}{p_B(\mathbf{x}_{i-1}|\mathbf{x}_i)}]$$

$$= -H(p_F(\mathbf{x}_0)) + \mathop{\mathbb{E}}_{\tau\sim p_F}[-\log p_B(\mathbf{x}_N) + \sum_{i=1}^{N}\log\frac{p_F(\mathbf{x}_i|\mathbf{x}_{i-1})}{p_B(\mathbf{x}_{i-1}|\mathbf{x}_i)}]$$

$$= -H(p_F(\mathbf{x}_0)) + \mathop{\mathbb{E}}_{\tau\sim p_F}[-\log p_B(\mathbf{x}_N) + \log\frac{g_0}{g_N} + \sum_{i=1}^{N}\log\frac{p(\delta_i^F)}{p(\delta_i^B)}].$$

The first and third terms are constant, so minimizing KL divergence is equivalent to minimizing

$$\mathop{\mathbb{E}}_{\tau\sim p_F}[-\log p_B(\mathbf{x}_N) + \sum_{i=0}^{N}\frac{1}{2}((\delta_i^B)^2 - (\delta_i^F)^2)].$$

We sample the trajectory $\tau$ from the forward process and $\delta_i^F$ is Gaussian random noise that is independent of the model. Thus, its expectation is a constant and the objective function can be simplified to

$$\mathop{\mathbb{E}}_{\tau\sim p_F}[-\log p_B(\mathbf{x}_N) + \sum_{i=1}^{N}\frac{1}{2}(\delta_i^B)^2]. \tag{23}$$

Additionally, for a given trajectory $\tau$, the following equations holds

$$f_i\Delta t + g_i\delta_i^F\sqrt{\Delta t} = [f_{i+1} - g_{i+1}^2s_{i+1}]\Delta t_i + g_{i+1}\delta_i^B\sqrt{\Delta t_i}.$$

Hence, the backward noise can be evaluated as

$$\delta_i^B = \frac{g_i}{g_{i+1}}\delta_i^F - [\frac{f_{i+1} - f_i}{g_{i+1}} + g_{i+1}s_{i+1}]\sqrt{\Delta t_i}. \tag{24}$$

# D Implemtation of Stochastic Adjoint in PyTorch [34]

Below is the implementation of stochastic adjoint method with `torch.autograd.Function` in PyTorch. The roles of most helper functions and variables can be informed from their names and comments.

```python
class SdeF(torch.autograd.Function):
    @staticmethod
    @amp.custom_fwd
    def forward(ctx, x, model, times, diffusion, condition,*m_params):
        shapes = [y0_.shape for y0_ in m_params]

        def _flatten(parameter):
            # flatten the gradient dict and parameter dict
            ...

        def _unflatten(tensor, length):
            # return object like parameter groups
            ...

        history_x_state = x.new_zeros(len(times) - 1, *x.shape)
        rtn_logabsdet = x.new_zeros(x.shape[0])
        delta_t = times[1:] - times[:-1]
        new_x = x
        with torch.no_grad():
            for i_th, cur_delta_t in enumerate(delta_t):
                history_x_state[i_th] = new_x
                new_x, new_logabsdet = model.forward_step(
                    new_x,
                    cur_delta_t,
                    condition[i_th],
                    condition[i_th + 1],
                    diffusion[i_th],
                    diffusion[i_th + 1],
                )
                rtn_logabsdet += new_logabsdet
        ctx.model = model
        ctx._flatten = _flatten
        ctx._unflatten = _unflatten
        ctx.nparam = np.sum([shape.numel() for shape in shapes])
        ctx.save_for_backward(
            history_x_state.clone(), new_x.clone(), times,
            diffusion, condition
        )
        return new_x, rtn_logabsdet

    @staticmethod
    @amp.custom_bwd
    def backward(ctx, dL_dz, dL_logabsdet):
        history_x_state, z, times, diffusion, condition = ctx.\
                                          saved_tensors
        dL_dparameter = dL_dz.new_zeros((1, ctx.nparam))

        model, _flatten, _unflatten = ctx.model, ctx._flatten, ctx.\
                                          _unflatten
        m_params = tuple(model.parameters())
        delta_t = times[1:] - times[:-1]
        with torch.no_grad():
            for bi_th, cur_delta_t in enumerate( \
                torch.flip(delta_t, (0,))):
                bi_th += 1
                with torch.set_grad_enabled(True):
                    x = history_x_state[-bi_th].requires_grad_(True)
                    z = z.requires_grad_(True)
                    noise_b = model.cal_backnoise(
                        x, z, cur_delta_t,
```

```
                condition[-bi_th], diffusion[-bi_th]
            )

            cur_delta_s = -0.5 * (
                torch.sum(noise_b.flatten(start_dim=1) ** 2,
                    dim=1)
            )
            dl_dprev_state, dl_dnext_state, *dl_model_b =
                torch.autograd.grad(
                    (cur_delta_s),
                    (x, z) + m_params,
                    grad_outputs=(dL_logabsdet),
                    allow_unused=True,
                    retain_graph=True,
                )
            dl_dx, *dl_model_f = torch.autograd.grad(
                (
                    model.cal_next_nodiffusion(
                        x, cur_delta_t, condition[-bi_th - 1]
                    )
                ),
                (x,) + m_params,
                grad_outputs=(dl_dnext_state + dL_dz),
                allow_unused=True,
                retain_graph=True,
            )
            del x, z, dl_dnext_state
        z = history_x_state[-bi_th]
        dL_dz = dl_dx + dl_dprev_state
        dL_dparameter += _flatten(dl_model_b).unsqueeze(0)
                    + _flatten(dl_model_f).unsqueeze(0)

    return (dL_dz, None, None, None, None,
            *_unflatten(dL_dparameter, (1,)))
```

# E    2-D sythetic examples

## E.1    Time discretization for DDPM

DDPM uses a fixed noising schema, $p(\mathbf{x}_i|\mathbf{x}_{i-1}) = \mathcal{N}(\mathbf{x}_i; \sqrt{1-\beta_i}\mathbf{x}_{i-1}, \beta_i\mathbf{I})$. The discretization can be reformulated as

$$\mathbf{x}_i = \sqrt{1-\beta_i}\mathbf{x}_{i-1} + \sqrt{\beta_i}\mathbf{w}_{i-1}.$$

Since $\beta_i$ is very small in DDPM implementation, the corresponding SDE is

$$d\mathbf{x} = -\frac{1}{2}\mathbf{x}dt + d\mathbf{w}, \tag{25}$$

and DDPM follows discretization $\Delta t_i = \beta_i$. Clearly, DDPM has a fixed forward diffusion $\mathbf{f}(\mathbf{x}, t) = -\frac{1}{2}\mathbf{x}$ with no learnable parameters. We can compare trajectory evolution of DDPM with that of DiffFlow in the following examples.

## E.2    Trajectories on 2D points

We use similar architectures for DiffFlow and DDPM. To capture the dependence of the drift as well as the score networks on time, we use the Fourier feature embeddings with learned frequencies. Points positions embeddings and the embedded time signals are added before standard MLP layers. In various points datasets, we use 3 layers MLP with 128 hidden neurons for DiffFlow. We increase the width for DDPM so that the two models have comparable sizes. We adopt the official code for evaluating FFJORD. To further illustrate the backward process of all those three models, we include the comparison in Figure 8. We use 30 steps sampling for DiffFlow and 500 steps for DDPM. It can be observed from Figure 8 that DiffFlow diffuses data efficiently and meanwhile keeps the details of the distributions better.

## E.3    NLL comparison on 2D points and Sharp Datasets

The negative log-likelihood results on 2-D data are reported in Table 6. We use `torchdiffeq`[3] for integral over time. We use `atol`$= 10^{-5}$ and `rtol`$= 10^{-5}$ for FFJORD and DiffFlow (ODE). Clearly, DiffFlow has

---

[3]`https://github.com/rtqichen/torchdiffeq`

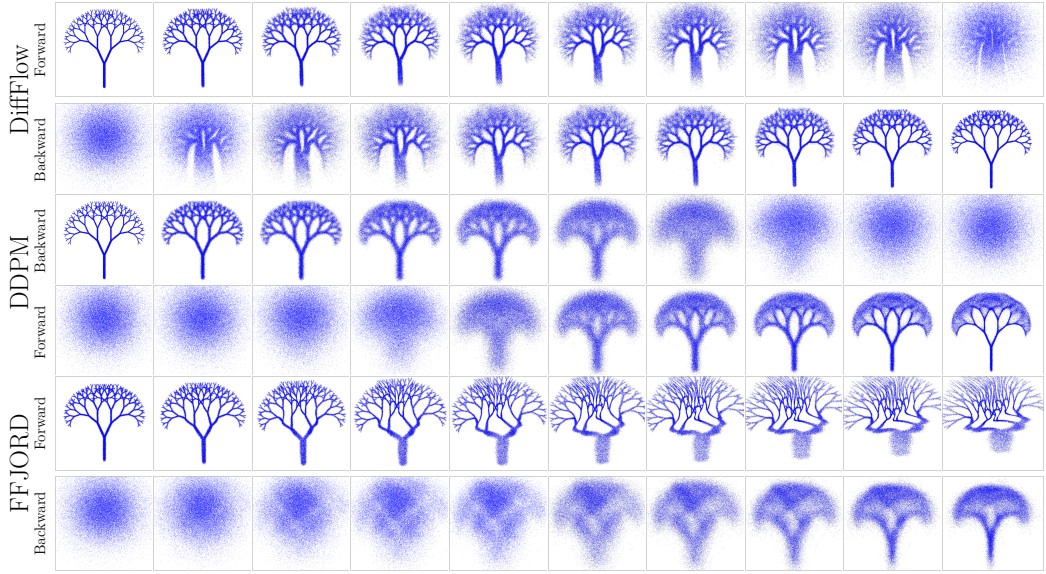

Figure 8: We compare the forward and backward sequences of DiffFlow, DDPM and FFJORD. Due to the unsatisfying forward process, the backward sequence and forward sequence for FFJORD are mismatched. The backward process of DDPM misses the details presented in the forward process. In contrast, DiffFlow has its backward trajectory distribution in alignment with the forward process.

.

much better performance on all these examples. The advantage of DiffFlow is even more obvious when the distribution concentrates on a 1d submanifold. Indeed, in the absence of noise as in FFJORD, it is easier to transform a smooth distribution than a sharp one to a Gaussian distribution. This phenomenon is better illustrated by comparing performance between Sharp Olympics and Olympics and their trajectories in Figure 9.

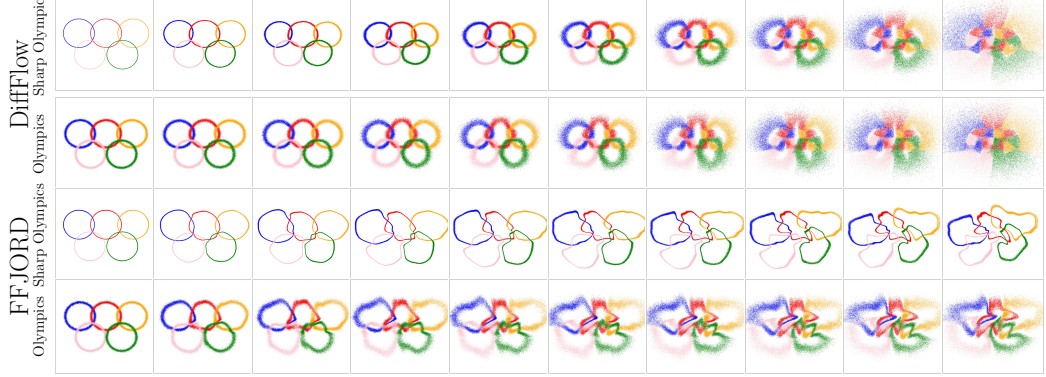

Figure 9: Forward sequences of DiffFlow and FFJORD on Sharp Olympics and Olympics. For FFJORD, it is more challenging to diffuse Sharp Olympics than Olympics. Thanks to the added noise, DiffFlow is more powerful in transforming data into Gaussian.

.

# F  Density estimation

We use similar network architectures with the following modifications. Since the ReZero [17] network shows better convergence property and often leads to lower negative log-likelihood, we use it instead of the MLP with the cost of additional number of layers parameters for ReZero. We find $N = 30$ works well for training on tabular datasets. We normalize the data with zero mean and standard deviation before feeding them into networks. We search over terminal time $T = 0.1, 0.25, 0.5, 0.75, 1.0$. We also try different widths and depths of the network. We stop training once the loss plateaus for five epochs. The architecture choices for different datasets can be found in Table 7. We rerun the density estimation with different random seeds and report the

Table 6: Negative log-likelihood on 2-d synthetic datasets.

| Dataset | RealNVP | FFJORD | CIF-ResFlow | DiffFlow (ODE) |
|---|---|---|---|---|
| Sharp Olympics | 2.12 | 1.52 | 1.32 | **-0.63** |
| Olympics | 2.19 | 1.64 | 1.59 | **1.24** |
| Checkerboard | 2.35 | 2.05 | 1.95 | **1.82** |
| Fractal Tree | 2.43 | 2.16 | 2.17 | **1.02** |
| Carpet | 2.47 | 2.37 | 3.32 | **2.08** |
| 2 Spirals | 2.04 | 1.84 | 1.83 | **1.73** |

| Dataset | layers | hidden dims | $T$ | batchsize |
|---|---|---|---|---|
| POWER | 3 | 128 | 1.0 | 20000 |
| GAS | 4 | 128 | 0.1 | 20000 |
| HEPMASS | 4 | 128 | 0.5 | 20000 |
| MINIBOONE | 3 | 128 | 0.75 | 2500 |
| BSDS300 | 3 | 256 | 0.1 | 20000 |

Table 7: DiffFlow architectures for density estimation on tabular data.

standard deviation over 3 runs in Table 8. DiffFlow is trained on one RTX 2080 Ti and the training takes between 14 minutes and 2 hours depending on the datasets and the model size.

## G  Image generation

### G.1  Settings

We rescale the images into $[-1, 1]$ for all experiments on images. We adopt U-net style model as used successfully in DDPM for both the drift and the score networks and they are trained with training data. We use the Adam optimizer with a learning rate $2 \times 10^{-4}$ for all three datasets. On MNIST, we use batchsize 128 and train 6k iterations. It takes 20 hours on one RTX 3090Ti. On CelebA, we train with $N = 10$ for 10k iterations, $N = 30$ for next 10k iterations and $N = 50$ for 10k iterations. It takes around 40 hours on 8 RTX 2080 Ti. On CIFAR10, we follow the similar scheduling over $N = \{10, 30, 50, 75\}$ for a total of 100k iterations. It takes around 4 days in 8 RTX 2080 Ti GPUs. We found that the exponential moving average (EMA) with a decay factor of 0.999 stabilizes the updating of the parameters of models. As it is reported in existing works [17, 41], we also found EMA dramatically improves the sampling quality. We save checkpoints for every 500 steps and report the best NLL and sampling quality among those checkpoints. We tested performance with three different random seeds and report mean in Table 2 and Table 3. The FIDs and NLL reported in this works are based on the selected checkpoints that achieve the best performance.

Among all hyperparameters, we find that the parameter $T$ is very crucial for training and sampling quality. The length of time interval deals with the trade-off between diffusion and learning backward process from the reconstruction. Larger $T$ injects more noise into the processes. On one hand, it helps diffuse data into Gaussian. On the other hand, larger $T$ requires backward process remove more noise to recover the original data or generate high-quality samples. Figure 10 shows the impacts of different $T$ values. We sweep the choice of $0.05, 0.1, 0.2, 0.5, 1.0$. $T = 0.05$ works best for CIFAR10 and $T = 0.1$ for CelebA.

| Dataset | POWER | GAS | HEPMASS | MINIBOONE | BSDS300 |
|---|---|---|---|---|---|
| RealNVP [8] | -0.17± 0.01 | -8.33± 0.14 | 18.71± 0.02 | 13.55± 0.49 | -153.28± 1.78 |
| FFJORD [14] | -0.46± 0.01 | -8.59± 0.12 | 14.92± 0.08 | 10.43± 0.04 | -157.40± 0.19 |
| MADE [11] | 3.08± 0.03 | -3.56± 0.04 | 20.98± 0.02 | 15.59± 0.50 | -148.85± 0.28 |
| MAF [33] | -0.24± 0.01 | -10.08± 0.02 | 17.70± 0.02 | 11.75± 0.44 | -155.69± 0.28 |
| TAN [31] | -0.48± 0.01 | -11.19± 0.02 | 15.12± 0.02 | 11.01± 0.48 | -157.03± 0.07 |
| NAF [18] | -0.62± 0.01 | -11.96± 0.33 | 15.09± 0.40 | 8.86± 0.15 | -157.73± 0.04 |
| DiffFlow (ODE) | -1.04± 0.01 | -10.45± 0.15 | 15.04± 0.04 | 8.06± 0.13 | -157.80± 0.17 |

Table 8: Average negative log-likelihood (in nats) on tabular datasets [33] for density estimation (lower is better).

## G.2 Image Completion

We also perform image completion tasks using our methods. The task is to complete the full images given partial masked images. The task is similar to image inpainting [27] but we use the whole masked forward trajectory instead of masked image [41, 17]. For a given real image $\mathbf{x}$, we denote the trajectories $\{\mathbf{x}_i^F\}_{i=0}^N$ by running forward process started with $\mathbf{x}_0 = \mathbf{x}$. For a given mask $\Gamma$, $\hat{\mathbf{x}}_i^F = \mathbf{x}_i^F \cdot \Gamma$ is the masked image, with $\cdot$ denotes element wise multiplication. $\{\hat{\mathbf{x}}_i^F\}_{i=0}^N$ gives information for backward process to complete original images. Instead of simulating reverse-time SDE, we incorporate the signals of $\{\hat{\mathbf{x}}_i^F\}_{i=0}^N$ by

$$\bar{\mathbf{x}}_{i-1}^B = \hat{\mathbf{x}}_i^B - [\mathbf{f}_i(\hat{\mathbf{x}}_i^B) - g_i^2 \mathbf{s}_i(\hat{\mathbf{x}}_i^B)]\Delta t_i + g_i \delta_i^B \sqrt{\Delta t_i} \tag{26a}$$

$$\hat{\mathbf{x}}_{i-1}^B = \bar{\mathbf{x}}_{i-1}^B \cdot (1 - \Gamma) + \hat{\mathbf{x}}_{i-1}^F. \tag{26b}$$

Compared with sampling through the backward process, the Equation (26) iteratively refine the masked regions based on the unmasked image regions. We present the image completion results in Figure 11 and Figure 12.

## G.3 More samples

We present more generated image samples in Figure 13, Figure 14 and Figure 15.

# H Marginal Equivalent SDEs

Below we provide a derivation for the marginal equivalent SDEs in Equation (16). The reverse time SDE in DiffFlow is

$$d\mathbf{x} = [\mathbf{f}(\mathbf{x}, t) - g^2(t)\mathbf{s}(\mathbf{x}, t)]dt + g(t)d\mathbf{w}, \tag{27}$$

where the drift $\mathbf{f} : \mathbb{R}^d \to \mathbb{R}^d$, the diffusion coefficients $g : \mathbb{R} \to \mathbb{R}$, and $\mathbf{w}(t) \in \mathbb{R}^d$ is a standard Brownian motion. According to Fokker-Planck-Kolmogorov (FPK) Equation for reverse time,

$$\frac{\partial p_1(\mathbf{x}, t)}{\partial t} = -\sum_i \frac{\partial}{\partial x_i}\{[f_i(\mathbf{x}, t) - g^2(t)s_i(\mathbf{x}, t)]p_1(\mathbf{x}, t)\} - \frac{g^2(t)}{2}\sum_i \frac{\partial^2}{\partial x_i^2}[p_1(\mathbf{x}, t)], \tag{28}$$

with $p_1(\mathbf{x}, t)$ being the marginal distribution induced by (27).

Now consider the SDE

$$d\mathbf{x} = [\mathbf{f}(\mathbf{x}, t) - \frac{1 + \lambda^2}{2}g^2(t)\mathbf{s}(x, t)]dt + \lambda g(t)d\mathbf{w} \tag{29}$$

with $\lambda \geq 0$. To simplify notation, we denote $\hat{\mathbf{f}}(\mathbf{x}, t) = \mathbf{f}(\mathbf{x}, t) - \frac{1+\lambda^2}{2}g^2(t)\mathbf{s}(x, t)$. According to Fokker-Planck-Kolmogorov (FPK) Equation, its marginal distribution evolves as

$$\frac{\partial p_2(\mathbf{x}, t)}{\partial t} = -\sum_i \frac{\partial}{\partial x_i}[\hat{f}_i(\mathbf{x}, t)p_2(\mathbf{x}, t)] - \frac{\lambda^2 g^2(t)}{2}\sum_i \frac{\partial^2}{\partial x_i^2}[p_2(\mathbf{x}, t)]. \tag{30}$$

Since

$$\sum_i \frac{\partial^2}{\partial x_i^2}[p(\mathbf{x}, t)] = \sum_i \frac{\partial}{\partial x_i}\frac{\partial}{\partial x_i}[p(\mathbf{x}, t)] = \sum_i \frac{\partial}{\partial x_i}[p(\mathbf{x}, t)\frac{\partial \log p(\mathbf{x}, t)}{\partial x_i}],$$

in view of $\mathbf{s}(\mathbf{x}, t) = \nabla \log p_2(\mathbf{x}, t)$, we can rewrite Equation (30) as

$$-\sum_i \frac{\partial}{\partial x_i}[\hat{f}_i(\mathbf{x}, t)p_2(\mathbf{x}, t)] - \frac{\lambda^2 g^2(t)}{2}\sum_i \frac{\partial^2}{\partial x_i^2}[p_2(\mathbf{x}, t)]$$

$$= -\sum_i \frac{\partial}{\partial x_i}[\hat{f}_i(\mathbf{x}, t)p_2(\mathbf{x}, t)] - \frac{(\lambda^2 - 1)g^2(t)}{2}\sum_i \frac{\partial}{\partial x_i}[p_2(\mathbf{x}, t)\frac{\partial \log p_2(\mathbf{x}, t)}{\partial x_i}] - \frac{g^2(t)}{2}\sum_i \frac{\partial^2}{\partial x_i^2}[p_2(\mathbf{x}, t)]$$

$$= -\sum_i \frac{\partial}{\partial x_i}\{[\hat{f}_i(\mathbf{x}, t) + \frac{\lambda^2 - 1}{2}g^2\frac{\partial \log p_2(\mathbf{x}, t)}{\partial x_i}]p_2(\mathbf{x}, t)\} - \frac{g^2(t)}{2}\sum_i \frac{\partial^2}{\partial x_i^2}[p_2(\mathbf{x}, t)]$$

$$= -\sum_i \frac{\partial}{\partial x_i}\{\left[f_i(\mathbf{x}, t) - g^2\frac{\partial \log p_2(\mathbf{x}, t)}{\partial x_i}\right]p_2(\mathbf{x}, t)\} - \frac{g^2(t)}{2}\sum_i \frac{\partial^2}{\partial x_i^2}[p_2(\mathbf{x}, t)].$$

Thus, given that $p_1(\mathbf{x}, T)$ and $p_2(\mathbf{x}, T)$ are identical, Equation (28) and Equation (30) are equivalent. Therefore, Equation (29) and Equation (27) share the same marginal distributions.

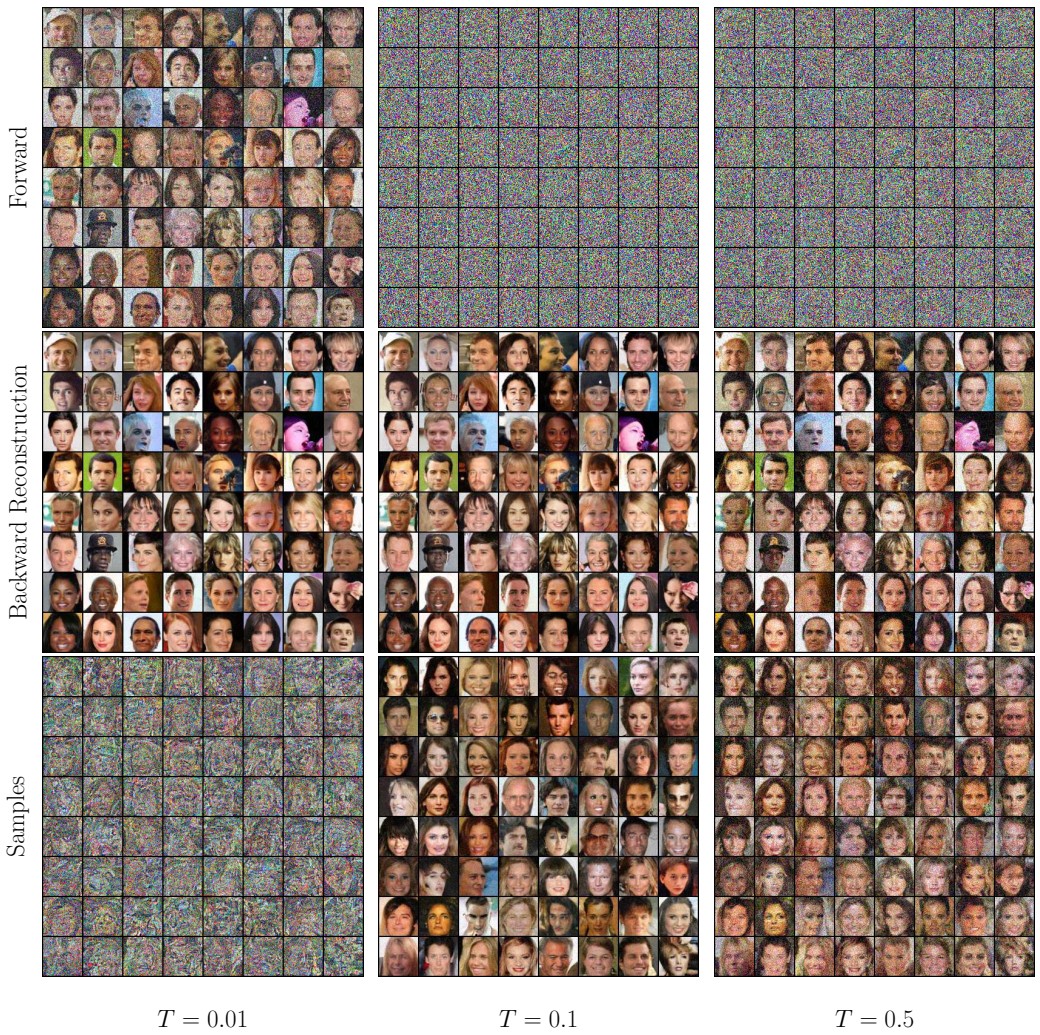

Figure 10: Illustration of impacts of time interval length $T$. First two rows show the trade-off between forward diffusion and backward reconstruction. The first row displays the $\mathbf{x}_N$ while the second row shows the reconstructed image from $\mathbf{x}_N$. We show generated samples from three models in the last row. With small $T$, it is easy to reconstruct origin data but difficult to diffuse the data. On the other hand, more noise makes it diffuse the data effectively but poses more challenges for reconstructions and as well as sampling high quality samples.

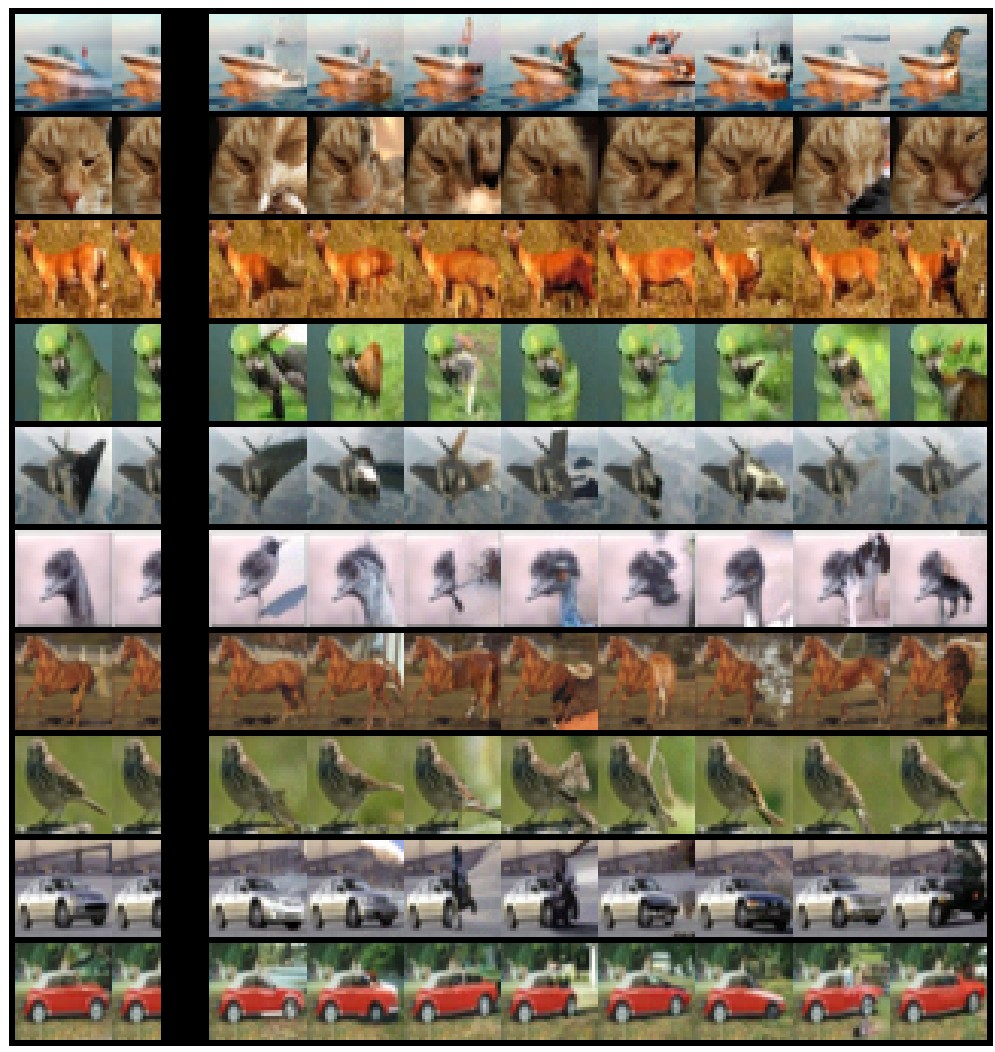

Figure 11: CIFAR10 image completion samples

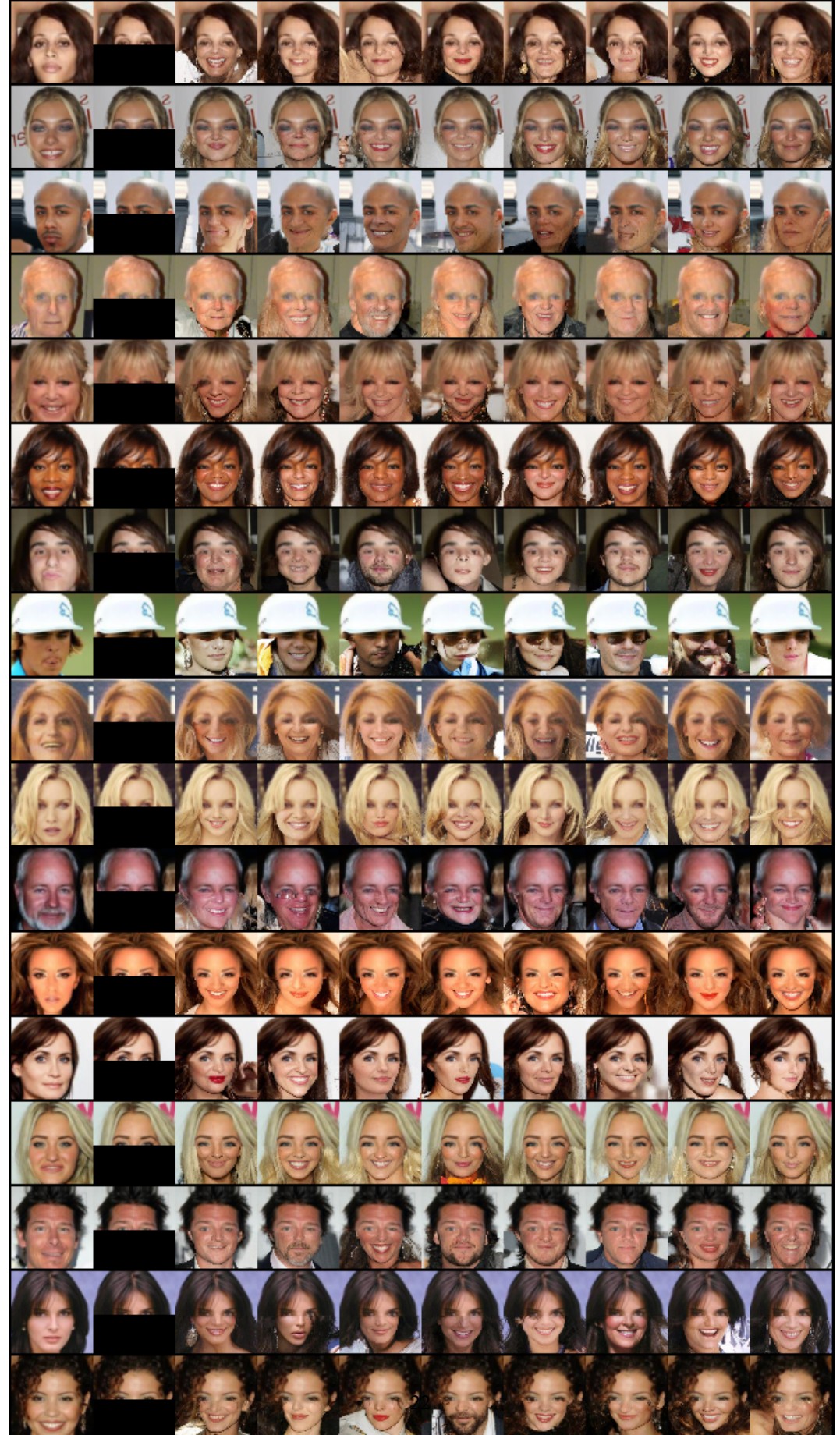

Figure 12: CelebA image completion samples

Figure 13: Uncurated MNIST Generated samples

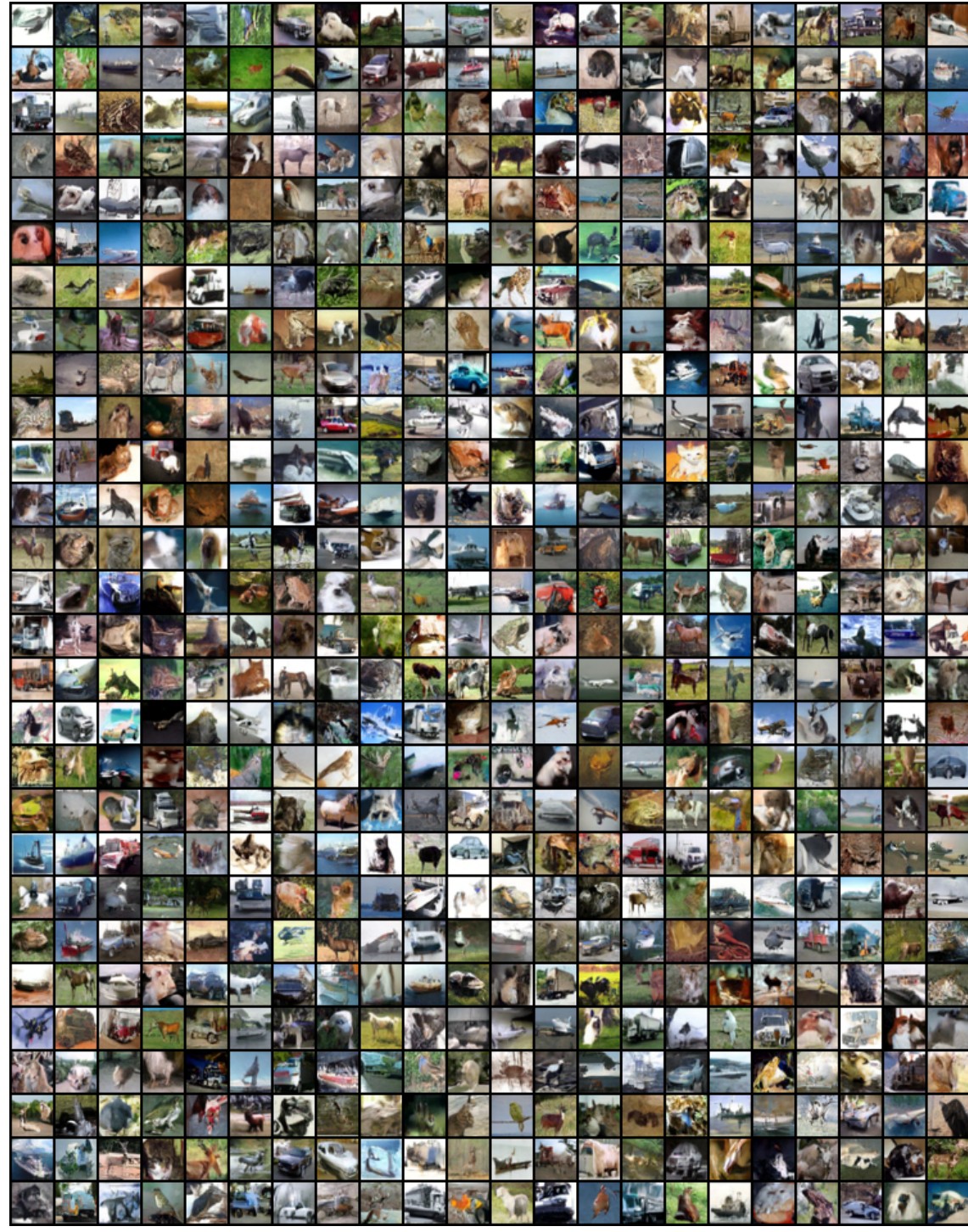

Figure 14: Uncurated CIFAR10 Generated samples

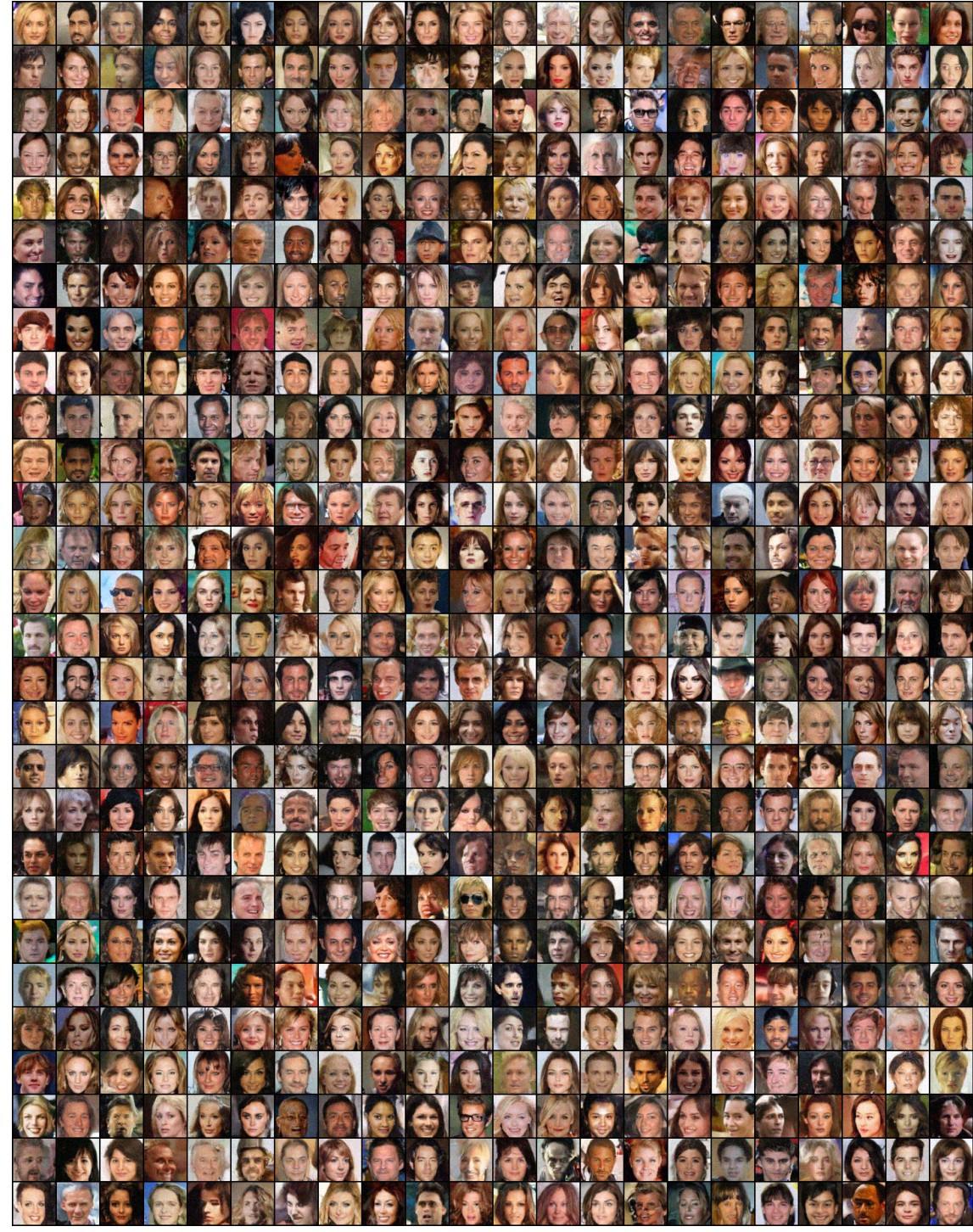

Figure 15: Uncurated CelebA Generated samples