# OpenReview forum: "Diffusion Normalizing Flow"
_NeurIPS.cc/2021/Conference — NeurIPS 2021 Poster_

### Official Review · Reviewer_i723 · 2021-07-16

**Rating:** 4
**Confidence:** 4

**Summary:**

This paper proposes "Diffusion Normalizing Flow", which generalizes current diffusion-based models to have a trainable inference process. This allows the model to transform structured data into an unstructured prior in a more efficient manner, thus resulting in faster sampling.

**Limitations And Societal Impact:**

See above.

**Main Review:**

### Significance and Limitation

The paper serves as a natural extension of [1] and [2], which have a fixed inference process that slowly transforms an arbitrary data distribution into an unstructured prior and learn to revert the dynamic. This paper proposes to parameterize the inference part so that the transformation of the data into an unstructured noise can be more efficient (as illustrated by Fig 4).

Two limitations of the proposed methods:

1. Computational time: Despite the generality this new framework provides, I suspect training will take much more time since one would have to backprop through the entire trajectory. This is in contrast to the proposal of a recent concurrent work [3], which, due to the constraint on the form of the inference process, still allows exact sampling of the perturbed data at any time step. I see this as a limitation since having to backprop through the entire trajectory means it will make it harder to scale up to larger datasets.
2. Memory: Backpropagation through the entire path also raises another problem — the increase in memory. This work proposes an "adjoint" method and claims to have constant memory consumption (in the number of integration steps). However, I found it to be not entirely true, as the algorithm requires caching all intermediate states $x_i$, which means it is still technically $O(N)$. A real constant-memory adjoint method would not require storing all the intermediate states, such as neural ODE, and another recent work [4], which suggests using the stochastic adjoint sensitivity method proposed by [5].

[1] Deep unsupervised learning using nonequilibrium thermodynamics

[2] Denoising diffusion probabilistic models

[3] Variational Diffusion Models

[4] A Variational Perspective on Diffusion-Based Generative Models and Score Matching

[5] Scalable gradients for stochastic differential equations

Presentation and clarity also need some work; see the following points for more detail.

### More technical questions and criticisms:

1. I don't think we can simply drop all the constants as claimed in L142. For example, the normalizing constants of the Gaussians will depend on the variance, which is the diffusion coefficients of the SDEs, unless these coefficients are either constant or not trained. Overall I find the derivation in Section 3.1 very confusing; perhaps having a detailed derivation in the appendix will help.
2. Is there a runtime comparison between DDPM and the proposed method in training? e.g. how much time does it take to make one parameter update, fixing the model size?
3. It is claimed that DiffFlow is a combination of flows and diffusion models (e.g. L179). It's not very clear to me how NF is integrated into this framework. Do you compose flows with diffusion models? Or it's just a matter of interpretation?
4. Aside from the use of the equivalent ODE for density evaluation (which is originally proposed by [6], is there other reasons to introduce the more general marginal-equivalent SDEs, i.e. eq (17)? Also, why not report the ELBO of the SDE?
5. What is the concept of "layer" within this paper? e.g. in L231, it says DiffFlow uses no more than 5 layers. Is it the size of the network that parameterizes f, g, and s?
6. The statement of Theorem 1 is very handwavy, and so is the proof. "is equivalent to", under what condition? In the proof it seems it requires fixing $p_B(x_i|x_{i-1})=p_F(x_i|x_{i-1})$, what does it mean by it's possible when g is small, e.g. a Gaussian approximation will suffice? It doesn't feel like a rigorous proof.

[6] Score-based generative modeling through stochastic differential equations

### Typos, and other suggestions:

- L36: I'd put "continuous-time" normalizing flows, instead of just normalizing flows
- L100: $f:\mathbf{R}^d\times \mathbf{R}\rightarrow \mathbf{R}^d$
- L103 $p_F$ is defined to be the distribution over trajectories, but then in L106 it's used as the score, which is the gradient of the marginal density. This is not accurate.
- Eq (10): why does it hold? Is it the data processing inequality? Please clarify.
- L192: noising transformation?
- In Appendix A, [39] is cited, but is not displayed in the paper.

**Time Spent Reviewing:**

5

---

> ### Author Response · Authors · 2021-08-10
> **Response to Reviewer i723**
>
> ### Q: Computational time
>
> The DiffFlow is a generative modeling framework that shares many features with both normalizing flow and diffusion models. To some degree, DiffFlow bridges diffusion models and normalizing flow. Thus we compare DiffFlow with both diffusion models and normalizing flows.
>
> As compared with most diffusion models such as DDPM, DiffFlow replaces the linear drift in the forward process with a trainable nonlinear drift. As a consequence, the conditional distributions $p(\mathbf x_i|\mathbf x_0), p(\mathbf x_{i-1}|\mathbf x_i, \mathbf x_0)$ of the forward process in DiffFlow do not have closed form and therefore the score matching training scheme is no longer applicable. Instead, we use a variational bound of the log-likelihood as the training objective. The cost function depends on the trajectories of the diffusion process and the training is thus more expensive. On the other hand, as a generalization of normalizing flow, the training time of DiffFlow is comparable to and often less than that of most normalizing flow.
>
> In addition to training time, the sampling time of generative models is also important. As illustrated in Figure 4,5,8,9, the flexible drift function in the learnable forward process of DiffFlow makes it possible to retain finer details of data distribution compared with DDPM, and thanks to the nonlinear forward diffusion, it takes much fewer sampling steps to achieve similar sampling quality.
>
> Finally, we note that [1] is a concurrent work.
>
> ### Q: Memory
>
> To train DiffFlow, we need to backpropagate through the (discretized) trajectories of the diffusion. In our stochastic adjoint method, we cache the intermediate state $\{\mathbf x_0,\mathbf x_1,\ldots,\mathbf x_N\}$ along the trajectories, requiring memory $\mathcal{O} (N)$. In contrast, the paper Scalable gradients for stochastic differential equations (referred to as SSDE later) solves the SDE backward to recover the intermediate states from the final states. To do so, one needs to cache the intermediate noises $d\mathbf w$ which also require memory $\mathcal{O} (N)$. SSDE further takes advantage of pseudo-random generator to save memory for the intermediate noises, reducing the memory requirements to $\mathcal{O} (1)$. Thus, SSDE reduces the memory requirements by solving the SDE backward with some extra amount of calculation. However, due to time discretization error, the backward process won't match exactly the forward one. This would introduce bias if used for the training of DiffFlow. One needs to use small time steps to reduce the discretization error but this would increase computational time. Our stochastic adjoint method doesn't have this issue since the exact intermediate states are cached and larger step size can be used. Based on this property, we propose progressively training trick (Section 3.2,3.3 and Appendix F1), which gradually increases $N$ for further speedup.
>
> Moreover, in practice, we observed that the memory used to cache the intermediate state $\{\mathbf x_0,\mathbf x_1,\ldots,\mathbf x_N\}$ is small compared with other part of the neural networks (less than 2 percent in our experiments). For instance, the runtime memory demanded by models and its gradient takes around 9GB while cached states of all 100 steps consume 0.15GB in CIFAR experiments.
>
> To summarize, SSDE consumes memory $\mathcal{O} (1)$ but requires a small discretization step. Our stochastic adjoint method stores the intermediate state $\{\mathbf x_0,\mathbf x_1,\ldots,\mathbf x_N\}$ and thus technically needs $\mathcal{O} (N)$ memory. However, in practice, this cache is really small compared with other part of the neural networks. We would like to emphasize that our method is different from naive backpropagation through the whole expanded computational graph. We don't store the values of the hidden layers. We will make our statement more precise in the revision.
>
> Finally, we note that [2] is a concurrent work.
>
> ### Q: Dropped constants in L142
>
> The derivation is given in Appendix B. We will improve it to make the derivation more clear. The constant being dropped is the normalizing constant associated with Gaussian random variable $\delta_i^B$. This can be safely dropped because $\delta_i^B$ is a zero-mean Gaussian variable with unit covariance (see (11),(12)).
>
> ### Q: Runtime comparision
>
> Empirically, we observed that DiffFlow is about 6 times slower than DDPM in 2d toy examples, 55 times slower in MNIST, 160 times slower without progressive training as discussed in Section 3.2,3.3 and Appendix F.1. Progressive training can speed up as much as 16 times faster in image datasets.
>
> Compared with the continuous normalizing flow with the same model architecture, DiffFlow takes almost the same amount of time for one iteration with fixing time step choice. To achieve similar performance, the training of DiffFlow is much faster than normalizing flows.
>
> ### Q: flows and diffusion models in L179
>
> DiffFlow generalizes normalizing flow by replacing the deterministic forward process in normalizing flow by one forward stochastic process and one backward process. Similar ideas have been used in [3]. Note that in deterministic normalizing flow the backward process is forced to be the inverse of the forward process and thus the two processes reduce to one. This requires the forward process to be bijective, which greatly restricts the expressive power of normalizing flow. In DiffFlow, we model the forward process with a diffusion and approximate its reverse-time counterpart with another diffusion. We use the Kullback-Leibler divergence between the two processes as the training cost; this cost turns out to be a variational lower bound of the log-likelihood. The DiffFlow is thus a stochastic generalization of normalizing flow and in fact, includes the standard normalizing flow as a limit case when the diffusion noise goes to 0 (see Theorem 1).
>
> ### Q: marginal-equivalent SDEs Eq. (13) and ELBO
>
> The more general marginal-equation SDEs (17) are used for evaluating ODE density and NLL. It can also be used for sampling with different levels of noise (different $\lambda$ values).
> Though in our experiments, we observed $\lambda=1.0$ leads to the best results.
>
> We did report ELBO.
> Entries $\text{DiffFlow} (L_\beta) \leq 3.71$ and $\text{DiffFlow} (\hat{L}_\beta) \leq 3.67$  in Table 3 are ELBO values. It is an ELBO value whenever it is a inequality in the NLL column.
>
> ### Q: the concept of "layer" within this paper
>
> 5 Layer means the layer of the networks for $f(t,x,\theta),s(t,x,\theta)$. $g(t)$ is not learnable.
>
> ### Q: Theorem 1
>
> Theorem 1 claims minimizing objective function (13) reduces to maximizing log-likelihood based on change of variable in normalizing flows when $g$ shrinks to zero. Minimizing (13) involves minimizing over $p_F$ and $p_B$. For a fixed $p_F$, minimizing over $p_B$ results in $p_F(x_i|x_{i-1})=p_B(x_i|x_{i-1})$ when $p_B$ has infinite expressive power. In DiffFlow, $p_B(x_{i-1}|x_i)$ is restricted to be a Gaussian, and thus $p_B(x_{i}|x_{i-1})$ cannot be equal to $p_F(x_i|x_{i-1})$ which is also assumed to be Gaussian. However, the difference between them can be made arbitrary small when $g$ is small. We will add these technical details in the revision.
>
> ### Q: Typos and suggestions
>
> We will correct the typos. $p_F$ denotes the distribution of the trajectory $p_F(\tau)$, the marginal distribution $p_F(\mathbf x_i)$, and the conditional distribution $p_F(\mathbf x_{i}|\mathbf x_{i-1})$. Sorry for the abuse of notation. We will clarify this in the revision. Equation (10) follows from the disintegration of measure theorem and the nonnegativity of KL divergence. More specifically, by disintegration of measure, $p_F(\tau) = \int p_F(\tau | \mathbf x(t)) p_F(\mathbf x(t)) d\mathbf x(t)$ and $p_B(\tau) = \int p_B(\tau | \mathbf x(t)) p_B(\mathbf x(t)) d\mathbf x(t)$. It follows that $KL(p_F(\tau)|p_B(\tau)) = KL(p_F(\mathbf x(t))|p_B(\mathbf x(t))) + \int KL(p_F(\tau |\mathbf x(t))|p_B(\tau|\mathbf x(t)))d\mathbf x(t)$ and thus (10).
> [39] should be [38]. We will update this reference in the revision.
>
> [1]. Variational Diffusion Models
>
> [2]. A Variational Perspective on Diffusion-Based Generative Models and Score Matching
>
> [3]. SurVAE Flows: Surjections to Bridge the Gap between VAEs and Flows

---

### Official Review · Reviewer_Sbrg · 2021-07-16

**Rating:** 6
**Confidence:** 3

**Summary:**

The paper under review introduce the Diffusion Normalizing Flow, which combines the diffusion model method and the normalizing flow method in their algorithm. The idea is clearly presented and the main part of the paper is focused on the experiment implementation of the algorithm.

**Limitations And Societal Impact:**

Negatives.

1. The model itself seems to lack theoretical justification. To be precise, The proposed diffusion model with equation (5), (6) does not come naturally from the cited references [34,16]. It seems that the original form of the backward diffusion process is first introduced in [37]. This is not clearly stated, if the reviewer did not miss other key references.

2. Furthermore, the score function $\mathbf s(x,t,\theta)$ appears in the backward process without derivation or justification. The one in [37] follows from the results in Anderson 1982.  The reviewer is curious about the original sources of equation (6), or can the author provides the derivation of this backward process.  The term "stochastic adjoint" is used in the paper, but the adjoint process is not well explained in the paper, together with its link to equation (6).

3. Once equation (5),(6) are introduced, the author then claim to consider the DiffFlow by letting $f(t,x)$ to be $f(t,x,\theta).$ Is there any reason to do so? Why the diffusion function $g(t)$ is fixed in this case. Does this change keep the property of the original backward process? From the reviewer's point of view, the algorithm is rather intuitive, is there any derivation of the backward or adjoint process here after $f(t,x,\theta)$ being used.

4. It seems that the algorithm is very similar to the recent paper [Efficient and Accurate Gradients for Neural SDEs]. Can the author do some comparison in this case? (e.g. memory, efficiency, etc..)



**Main Review:**

Positive.

1. The idea seems to be new. Indeed, the proposed DiffFlow is one type of the Neural SDEs, where the volatility only depends on time variable.

2. The idea and the comparison to diffusion model and normalizing flow is clearly presented in the figure and one theoretical result is provided to link the three different methods.



**Time Spent Reviewing:**

8

---

> ### Author Response · Authors · 2021-08-10
> **Response to Reviewer Sbrg**
>
> Thank you for the detailed review and thoughtful feedback. Below we address specific questions.
>
> ### Q: Clarify Equation (5), (6) and score function $s(\mathbf x,t,\theta)$
>
> Equation (5) is a standard diffusion process and Equation (6) is a reverse-time diffusion. When the function $s(\mathbf x,t,\theta)$ in (6) is equal to the score function $\nabla \log p_F$, the reverse-time diffusion (6) induces the same distribution over the path space as the forward process (5). This is a classical result in diffusion processes and it has been used in [37]. The formula dates back to "Dynamical theories of Brownian motion" (Chapter 13) by Nelson in 1967 and Anderson 1982. We will add these references. We will also add a section in the Appendix to introduce diffusion processes and stochastic calculus.
>
> ### Q: Stochastic adjoint
>
> The adjoint process is $\frac{\partial L}{\partial \mathbf x_{i}}$ and it is not directly related to (6). The name "adjoint process'' follows Neural ODE and optimal control theory.
>
> ### Q: $f(\mathbf x,t)$ to $f(\mathbf x,t,\theta)$
>
> The only difference between $f(\mathbf x, t, \theta)$ and $f(\mathbf x,t)$ is that $f(\mathbf x, t, \theta)$ has learnable parameters $\theta$ that are to be determined through training. For instance, $f(\mathbf x, t) = -\mathbf x$ and $f(\mathbf x, t, \theta)= -\theta\mathbf x$. For any fixed parameters, the diffusion with drift $f(\mathbf x, t, \theta)$ works the same way as the diffusion with drift $f(\mathbf x, t)$. In the original diffusion model, the drift $f(\mathbf x, t)$ is hand-crafted and the forward process is restricted to be a special linear diffusion. In DiffFlow, we make $f(\mathbf x, t, \theta)$ trainable so that it can transform data into Gaussian noise more efficiently, resulting in sampling with less steps eventually. The advantage of a trainable $f(\mathbf x, t, \theta)$ is illustrated by Figure 4, 5. DiffFlow learns data-dependant noising process and models density with sharp details.
> The noise intensity $g(t)$ can also be made trainable as $g(t,\theta)$ with no extra difficulty. All the steps in the paper would carry through.
>
>
> ### Q: Comparison to Efficient and Accurate Gradients for Neural SDEs
>
> First of all, this is a concurrent work (referred to as NSDE later). Second, the motivations of this work and DiffFlow are quite different. In DiffFlow, we use learnable SDEs to learn generative models. In contrast, NSDE aims to develop more efficient algorithms for the training of SDEs.
>
> Finally, the method we developed to train SDEs for DiffFlow is different from that in NSDE. In our stochastic adjoint method, we cache the intermediate state $\{\mathbf x_0,\mathbf x_1,\ldots,\mathbf x_N\}$ along the trajectories, requiring memory $\mathcal{O} (N)$. In contrast, NSDE uses a reversible Heun solver to recover the intermediate states from the final states. To do so, one needs to cache the intermediate noises $d\mathbf w$ which also require memory $\mathcal{O} (N)$. NSDE further takes advantage of the pseudo-random generator to save memory for the intermediate noises, reducing the memory requirements to $\mathcal{O} (1)$. Thus, NSDE reduces the memory requirements by solving the SDE backward with some extra amount of calculation. In practice, we observed that the memory used to cache the intermediate state $\{\mathbf x_0,\mathbf x_1,\ldots,\mathbf x_N\}$ is small compared with memory of the neural networks (less than 2 percent in our experiments).

---

### Official Review · Reviewer_H9qy · 2021-07-19

**Rating:** 4
**Confidence:** 4

**Summary:**

This paper proposes to lean a more expressive forward process in a diffusion model by using a neural network drift function.

**Limitations And Societal Impact:**

Limitations of the paper are not adequately addressed, see the main review.

**Main Review:**

I think it is a nice insight that diffusion models can be thought of as relaxed normalizing flows. The method to learn a more expressive forward process is new, although it is straightforward and does not really resolve key issues in learning more expressive forward processes. For example, it is well known that the affine forward processes allow for sampling of $q(x_t | x_0)$ for arbitrary $t$, allowing for fast training by picking random $t$; furthermore, variance of training and evaluation can be reduced by Rao-Blackwellizing the variational lower bound into $x_0$ MSE prediction losses. Both properties do not seem true with the presented method when f is a neural network, and the computational complexity tradeoffs should be studied more thoroughly (e.g. what is the quantitative benefit of slower training time vs better sample quality or log likelihood?)

While it is interesting to learn more expressive forward processes, it is not clear why the method needs to be presented as a combination of diffusion and normalizing flows: it seems clear enough to present the method as learning a more expressive forward process for diffusion models. Also, I found it difficult to understand the description of time discretization in Section 3.3.

In Section 4.1, I am surprised that DDPM "blurs density details." Shouldn't it be possible to resolve this by carefully tuning the noise schedule? What hyperparameters were used for this experiment?

I find Table 4 slightly misleading. It is true that the FID scores are better than DDPM for small N, but for large N, the FID scores of DDPM are around 3, and it seems that the presented method does not approach that score.

Why does the table omit log likelihoods and FID scores from published methods that perform better, e.g. from Score-Based Generative Modeling through Stochastic Differential Equations by Song et al? The paper claims state-of-the-art NLL (line 246), but it is not.

The main limitation of the paper is that as it is presented, the results do not significantly improve upon the baseline enough to warrant the more complex training procedure and model construction, there is no adequate study of the training time (or FLOPS) vs NLL or sample quality, and in some cases, the results are presented in a slightly misleading way (e.g. Table 4).

**Time Spent Reviewing:**

2

---

> ### Author Response · Authors · 2021-08-10
> **Response to Reviewer H9qy**
>
> Thank you for the detailed review and thoughtful feedback. Below we address specific questions.
>
> ### Q: Pros and cons when $f$ is a neural network.
>
> The DiffFlow is a generative modeling framework that shares many features with both normalizing flow and diffusion models. Any algorithm has pros and cons, so is DiffFlow.
>
> As compared with most diffusion models such as DDPM, DiffFlow replaces the linear drift in the forward process with a trainable nonlinear drift. As a consequence, the conditional distributions $p(\mathbf x_t|\mathbf x_0), p(\mathbf x_{t-1}|\mathbf x_t, \mathbf x_0)$ of the forward process in DiffFlow do not have closed form and therefore the score matching training scheme is no longer applicable. Instead, we use a variational bound of the log-likelihood as the training objective. The cost function depends on the trajectories of the diffusion process and the training is thus more expensive. Empirically, we observe that DiffFlow has a better FID score than DDPM~($L$) which is trained by maximizing log-likelihood bound, but worse than DDPM($L_{simple}$). However, DiffFlow achieves better performance in terms of NLL than most diffusion models including DDPM and its variants with similar network size and architecture.
> As illustrated in Figure 4,5,8,9, the flexible drift function in the learnable forward process makes it possible to retain finer details of data distribution compared with DDPM. Moreover, thanks to the nonlinear forward diffusion, it takes much fewer sampling steps to achieve similar sampling quality.
>
> DiffFlow generalizes normalizing flow by replacing the deterministic forward process in normalizing flow by one forward stochastic process and one backward process. Note that in deterministic normalizing flow the backward process is forced to be the inverse of the forward process and thus the two processes reduce to one. This requires the forward process to be bijective, which greatly restricts the expressive power of normalizing flow. In DiffFlow, we model the forward process with a diffusion and approximate its reverse-time counterpart with another diffusion. We use the Kullback-Leibler divergence between the two processes as the training cost; this cost turns out to be a variational lower bound of the log-likelihood. The DiffFlow is thus a stochastic generalization of normalizing flow and in fact includes the standard normalizing flow as a limit case when the diffusion noise goes to 0 (see Theorem 1).
> Empirically, we observed that DiffFlow generates much better samples (in terms of both visualization and NLL) in some popular datasets than almost all existing normalizing flows. Moreover, the training of DiffFlow is often faster than normalizing flow as smaller discretization step $N$ is needed for DiffFlow.
>
> ### Q: Diffusion models and normalizing flows
>
> The DiffFlow is a generative modeling framework that shares many features with both normalizing flow and diffusion models. To some degree, DiffFlow bridges diffusion models and normalizing flow. This is clearly illustrated in Figure 4, from which we see that the way the trajectories of DiffFlow evolve shares some similarities with both diffusion models and normalizing flows. Compared with diffusion models, DiffFlow uses a trainable forward process. Compared with normalizing flow, DiffFlow uses two trainable diffusion processes instead of one deterministic forward process and uses a variational lower bound instead of the exact log-likelihood in normalizing flow. In fact, the standard normalizing flow can be viewed as a limit of DiffFlow when the intensity of noise in the diffusion goes to zero, as described in Theorem 1. Thus, we feel it is important and beneficial to inform researchers in both areas of the development of DiffFlow.
>
> ### Q: Time discretization in Section 3.3
>
> We proposed two time discretization schemes for the training of DiffFlow, fixed timestamps $L_\beta$ and flexible timestamps $\hat{L}_\beta$. In $L_\beta$, the time interval $[0,T]$ is discretized at the points $\{t_0, t_1, \ldots,t_N\}$ with $t_i = (\frac{i}{N})^\beta T$. The coefficient $\beta$ is set to be $0.9$ is our experiments. In $\hat{L}_\beta$, we train different batches with different time discretizaitons points, where the discretization point $t_i$ is sampled uniformly from the interval $[(\frac{i-1}{N-1})^\beta T, (\frac{i}{N-1})^\beta T]$.
> We observed that the latter has better scalability and stability when progressively increasing $N$ during training.
>
> ### Q: DDPM "blurs density details"
>
> We used the official code for [DDPM](https://github.com/hojonathanho/diffusion).
> The hyperparameters, noising scheduling $\beta_i,\alpha_i$ ,follow DDPM paper with number of denoising step being $N=500$ or $N=1000$. Both DiffFlow and DDPM use the same time condition MLP networks.
> We try both linear scheduling and cosine scheduling as suggested in [1] for the noising scheduling. DiffFlow is trained with 25 epochs while DDPM is trained with 300 epochs; both take around 30 mins with one Nvidia RTX 2080Ti GPU. We observe in Figure 5 that DiffFlow learns finer details than DDPM, especially on the sharp Olympics, the fractal tree and Sierpinski carpet datasets. We believe one reason that DDPM blurs density details is that the forward linear diffusion in DDPM adds noise to all the data points in the same fashion as illustrated in Figure 4. This noise mixes different parts of the data distribution that are close to each other (e.g., the intersection area of the Olympic rings) too quickly so that it is difficult to remove the noise to recover these details. In contrast, the learnable forward process in DiffFlow makes it possible to add noise slowly to these areas with finer details.
>
> ### Table 4 misleading
>
> Yes, DDPM($L_{simple}$) has better performance in terms of FID and this is already illustrated in Table 3. The purpose of Table 4 is to investigate the sensitivity of the performance of DiffFlow with respect to the value of $N$. We will clarify this in the revision.
>
> ### Comparison to existing algorithms
>
> We compared the performances of DiffFlow with normalizing flows and diffusion models. As can be seen from the following table, DiffFlow achieves a better FID score than normalizing flows and competitive performance compared with DDPM trained with unweighted variational bounds, DDPM $(L)$ and Improved DDPM. But it is worse than DDPM trained with reweighted loss, DDPM $(L_{simple})$, DDPM cont, and DDPM++. We can see that DiffFlow has the lowest relative FID degeneracy ratio and retains the best performance when the number of step is small ($N=5$). We believe sample quality can be further improved with a better choice of time discretization and reweighted loss function.
> In terms of NLL, DiffFlow performs better than normalizing flow and most diffusion models (when this paper was written). We list below the NLL for several representative algorithms. The score-based methods (DDPM cont.) have NLL 3.21/3.05. DiffFlow is slightly worse than DDPM++ (sub-VP, deep, sub-VP) and Improved DDPM, however, DDPM++ and Improved DDPM conduct multiple architectural improvements and use much deeper and wider networks.
>
> **NLLs of DiffFlow, DDPM and its variants. NLLs are measured in bits/dim. DDPM\* are reported based on variational lower-bound.**
>
> |Algorithm | NLL| CIFAR10 FID|
> |-|-|-|
> |DDPM* ($L$)[4] | $\leq 3.70$| 13.51|
> |DDPM* ($L_{simple}$)[4] | $\leq 3.75$| 3.17|
> |DiffFlow($\hat{L}_{ode}$) | 3.04| 14.14 |
> |DiffFlow($\hat{L}_\beta$) | $\leq 3.67$ | 13.43 |
> |DiffFlow($L_\beta$) | $\leq 3.71$| 13.87 |
> |DDPM[2] | 3.28| 3.37 |
> |DDPM cont. (VP)[2] | 3.21| 3.69 |
> |DDPM cont. (sub-VP)[2] | 3.05| 3.56 |
> |DDPM++ (VP)[2] | 3.16| 3.93 |
> |DDPM++ (sub-VP)[2] | 3.02| 3.16 |
> |DDPM++ (deep, VP)[2] | 3.13| 3.08 |
> |DDPM++ (deep, sub-VP)[2] | 2.99| 2.92|
> |Improved DDPM[3] | 2.94| 11.47|
>
> **FIDs with various $N$**
>
> |$N$ | DiffFlow | DDPM ($L$) | DDPM ($L_{simple}$) | DDPM cont. (sub-VP)|
> |-|-|-|-|-|
> |5   | 28.31 | 373.51 | 370.23  | 371.01    |
> |10  | 22.56 | 364.64 | 365.12  | 363.90    |
> |20  | 17.98 | 138.84 | 135.44  | 124.52    |
> |50  | 14.72 | 47.12  | 34.56   | 33.23     |
> |100 | 13.43 | 22.23  | 10.04   | 10.00     |
>
> **Relative FIDs degeneracy ratio, compared with  $N=100$**
>
> |$N$ | DiffFlow | DDPM ($L$) | DDPM ($L_{simple}$) | DDPM cont. (sub-VP)|
> |----|----------|------------|---------------------|---------|
> |5   | 1.77 | 16.80 | 37.02  | 37.10  |
> |10  | 1.40 | 16.40 | 36.12  | 36.39  |
> |20  | 1.18 | 6.24  | 13.54  | 12.45  |
> |50  | 1.09 | 2.12  | 3.45   | 3.23  |
> |100 | 1.0  | 1.0   | 1.0    | 1.0  |
>
> ### Q: Adequate study of the training time (or FLOPS) vs NLL or sample quality
>
> We will add curves of the NLL and sample quality vs training time in the appendix.
>
> ### Note
>
> During comparing FID scores, we find score from [torch-fidelity](https://github.com/toshas/torch-fidelity) differs from our original implementation. Thanks to its popularity and precision, we switch to torch-fidelity implementation. We reselect the best model among saved checkpoints and update FID scores of DiffFlow. We will update them in the revision. For experiments of relative FIDs degeneracy ratio, official codebase and checkpoints, [DDPM](https://github.com/hojonathanho/diffusion) and [DDPM cont. (sub-VP)](https://github.com/yang-song/score_sde_pytorch) are used to generate image samples.
>
> [1]. Improved denoising diffusion probabilistic models
>
> [2]. Score-Based Generative Modeling through Stochastic Differential Equations
>
> [3]. Improved denoising diffusion probabilistic models
>
> [4]. Denoising diffusion probabilistic models

---

### Official Review · Reviewer_b7rB · 2021-07-21

**Rating:** 8
**Confidence:** 5

**Summary:**

The paper introduces a non-linear drift component to the SDE formulation of diffusion-based generative modelling. Earlier work (DDPM) used a fixed linear drift component. This paper uses a trainable neural network as the drift component, and another one to estimate the score. Both these neural networks are trained by minimizing the KL divergence between the forward and backward trajectories, while accounting for their SDE formulations. It is found that this trained network performs reasonably well compared to prior works, and achieves better sample quality and shorter trajectories compared to DDPM as well as Continuous Normalizing Flows (FFJORD).

**Limitations And Societal Impact:**

The appendix describes the computational resources used in training the models. The figures and tables clearly illustrate the limitations of the method. It could be mentioned that lower sampling steps could alleviate computational burden significantly.

**Main Review:**

The paper is clear enough to understand, although there are some typos in a few places (mentioned below).

The "novel" contribution of making the drift component non-linear is relevant in the context of this line of research : discrete Markov chains (DDPM [1], etc.), to equivalent continuous formulations with the help of SDEs [2], to a variety of SDEs where the drift component is learnt as well instead of being prefixed.

Relevant examples have been shown in 2D data as well as image sampling to highlight the advantages of this method compared to the previous version (DDPM). Although the continuous formulation (SDE) is mentioned, the paper converts it back to a discrete Markov chain and trains it accordingly.

Figures 5, 8 and 9 illustrate the relevance of the current method compared to DDPM and FFJORD. They make a convincing argument for the use of a trainable forward process. Lines 182-184 make a convincing argument for why this particular type of 2D data was chosen. It is all the more encouraging that even though finer details are better, it is not at the expense of likelihood/NLL (in many cases).

A major significance is that the number of sampling steps can be quite low compared to the original formulation (Table 4) and yet the quality of generated samples does not significantly deteriorate as much as the baseline (DDPM).

Table 3 compares the NLL and FID of previous relevant methods for CIFAR10. DDIM [3] is missing from the list, as it is also a discrete formulation like DDPM but uses a different sampling scheme. It is a relevant comparison since DDIM also aims to reduce the number of training steps. Although, the method proposed is significantly different from the motivation of DDIM, in that DDIM preserves the DDPM formulation and only changes the sampling scheme, while the paper changes the forward process itself.

The appendix mentions the resources used to complete each experiment. It needs to be seen whether computational efficiency could be improved.

Code has been provided to reproduce the experiments.

[1] Denoising Diffusion Probabilistic Models, NeurIPS 2020
[2] Score-based generative modeling through stochastic differential equations, ICLR 2021
[3] Denoising Diffusion Implicit Models, ICLR 2021

Line 87 says x(0) = z and x(T) = x, but lines 121-122 say x(T) = x and x(0) = z. I assume the latter, it is good if this is clarified.

Line 55 : the forward process from x* to z* ?

Line 166 : (to add) in Figure 3*

Line 192 : noi*sing

Line 198 : to*pological?

Figure 5 : Frac*tal tree

Line 239 : using*

---

Update : I stand by my original evaluation of accepting this paper:

The "novel" contribution of making the drift component non-linear is relevant in the context of this line of research
I found their methodology to be well motivated
The pros and cons of each design step have been mentioned
Enough experiments have been performed to explore the method, with the pros and cons
Code has been provided for reproduceability
Computational resources used for each experiment have been mentioned
Reviewers' comments have been responded to sufficiently and honestly
If the other reviewers feel there are still some unanswered questions, or comments that have not been responded to, please let us know. I think this paper deserves to be accepted.

**Time Spent Reviewing:**

5

---

> ### Author Response · Authors · 2021-08-10
> **Response to Reviewer b7rB**
>
> We appreciate your time and patience. We will fix the typos and polish the paper.
>
> ### DDIM
>
> The two methods, DDIM and DiffFlow, to reduce sampling time in diffusion models are quite different. The training of DDIM is the same as that of DDPM. For sampling, a well-crafted non-Markovian diffusion process is constructed in DDIM to enable sampling with fewer discretization steps. In contrast, the forward process in DiffFlow is a learnable nonlinear diffusion. The training cost is a variational lower bound of the log-likelihood, which cannot be converted to the score matching cost as in DDPM/DDIM due to the nonlinear forward drift.
> DDIM achieves FID score 4.12 when $N=1000$ (image samples are generated based on official codebase and checkpoints), better than DiffFlow. The NLL for DDIM is not available. We present the FID scores for DDIM, DDPM, and DiffFlow with different sampling steps $N$ in the following two tables. We can see that DiffFlow has the lowest relative FID degeneracy ratio and retains the best performance when the number of step is small ($N=5$). We believe DiffFlow can be further improved with a better choice of time discretization and reweighted loss function. In addition to reducing sampling steps, DiffFlow uses a learnable forward nonlinear diffusion to add noise to the data, capturing finer details of the data distribution than DDPM as illustrated in Figure 4 and Figure 5.
>
> **FIDs with various $N$**
>
> |$N$ | DiffFlow | DDPM ($L$) | DDPM ($L_{simple}$) | DDIM    |
> |----|----------|------------|---------------------|---------|
> |5   | 28.31   | 373.51     | 370.23              | 44.69   |
> |10  | 22.56   | 364.64     | 365.12              | 18.62   |
> |20  | 17.98   | 138.84     | 135.44              | 10.89   |
> |50  | 14.72   | 47.12      | 34.56               | 7.01    |
> |100 | 13.43   | 22.23      | 10.04               | 5.63    |
>
> **Relative FIDs degeneracy ratio, compared with  $N=100$**
>
> |$N$    | DiffFlow  | DDPM ($L$) | DDPM ($L_{simple}$) | DDIM   |
> |----|----------|------------|---------------------|---------|
> |5      |  2.12  | 16.80      | 37.02               | 7.94   |
> |10     |  1.68  | 16.40      | 36.12               | 3.31   |
> |20     |  1.34  | 6.24       | 13.54               | 1.93   |
> |50     |  1.10  | 2.12       | 3.45                | 1.24   |
> |100    |  1.00  | 1.0        | 1.0                 | 1.0    |
>
>
> During comparing FID scores, we find score from [torch-fidelity](https://github.com/toshas/torch-fidelity) differs from our original implementation. Thanks to its popularity and precision, we switch to torch-fidelity implementation. We reselect the best model among saved checkpoints and update FID scores of DiffFlow. We will update them in the revision.
>
> ### Computational efficiency
>
> Unlike DDPM, the conditional distributions $p(\mathbf x_i|\mathbf x_0), p(\mathbf x_{i-1}|\mathbf x_i, \mathbf x_0)$ of the forward process in DiffFlow do not have closed form. To evaluate the gradient, DiffFlow needs to backpropagate through the trajectories and is thus significantly slower than DDPM for training. Empirically, we observe that DiffFlow is about 6 times slower than DDPM in 2d toy examples, 55 times slower in MNIST, 160 times slower in CIFAR10 without progressive training as discussed in Section 3.2, Section 3.3, and Appendix F.1. Progressive training can speed up training up to 16 times.
> Possible approaches to further improve the computational efficiency of DiffFlow include adding regularizers of the nonlinear drift and score network so that the discretization of SDEs can be made coarser, more expressive time condition networks with small $N$, better Neural SDE solvers, and better progressive training scheduling, etc.

---

### Decision · Program_Chairs · 2021-09-27

**Decision:**

Accept (Poster)

**Comment:**

There was significant discussion of this paper, with one reviewer increasing their score. The score range was unusually wide, even after discussion.

Reviewers felt that the idea presented was both novel and interesting. I agree with this assessment -- I think this paper draws a very nice and interesting link between normalizing flows and diffusion models. It's impressive to cast them into the same theoretical framework, and I think this will inspire followup work.
However, reviewers also had remaining concerns. Most seriously that:
- the computational tradeoffs during training that this approach required weren't clearly acknowledged or discussed. I believe this is a fair criticism. Please include discussion of this, and the training time tradeoffs you discussed in your rebuttal, in any camera ready.
- that the memory discussion around the adjoint method is misleading. I am unable to fully judge this concern. I do think something like logarithmic checkpointing should also be able to prevent memory overhead from being an issue though, so it's not clear to me that memory usage was ever a critical problem to start with.
- that the performance table left out a number of results with better FID and log likelihood scores than the presented method. This is definitely true, and the additional rows **must** be added for any camera ready. The paper only claimed "competitive performance", not SOTA, so I do not believe this is a fatal flaw.
- there were also some concerns about clarity, and I encourage the authors to address the specific aspects that reviewers found challenging.

Due to these legitimate concerns, I struggled with my recommendation for this paper, and read the paper myself to make a more informed decision. Because of the novelty of the approach, and because of my judgement of the relative importance of novelty to scientific progress, I am recommending acceptance.